# Real-time monitoring polymerization degree of organic photovoltaic materials toward no batch-to-batch variations in device performance

Lin-Yong Xu[1,2], Wei Wang[1,2], Xinrong Yang[1], Shanshan Wang[1], Yiming Shao[1], Mingxia Chen[1], Rui Sun [1] ✉ & Jie Min [1] ✉

Polymerization degree plays a vital role in material properties. Previous methodologies of molecular weight control generally cannot suppress or alleviate batch-to-batch variations in device performance, especially in polymer solar cells. Herein, we develop an in-situ photoluminescence system in tandem with a set of analysis and processing procedures to track and estimate the polymerization degree of organic photovoltaic materials. To support the development of this protocol, we introduce polymer acceptor PYT constructed by near-infrared Y-series small molecule acceptors *via* Stille polymerization, and shed light on the correlations between molecular weight, spectral parameters, and device efficiencies that enable the design of the optical setup and confirm its feasibility. The universality is verified in PYT derivatives with stereoregularity and fluoro-substitution as well as benzo[1,2-b:4,5-b']dithiophene-based polymers. Overall, our result provides a tool to tailor suitable conjugated oligomers applied to polymer solar cells and other organic electronics for industrial scalability and desired cost reduction.

Owing to their tremendous conducting properties conjugated polymers have been utilized in a broad of applications and continue to revolutionize the field of printed and flexible organic electronics[1,2], such as organic light-emitting diodes[3,4], organic field-effect transistor[5,6], and polymer solar cells (PSCs)[5,7,8] and so on[9–11]. Aside from the molecular structure and functional groups, an increasing number of chemists and materials scientists have recognized the pivotal role of the degree of polymerization (DP) of conjugated polymers in determining device performance[1,12,13]. Correlations between DP and material properties (including photoelectrical property, solution processability, crystallization behavior, flexibility, and morphological phase, etc.[1,2,12,14–19]) are ubiquitous in all kinds of synthetic conjugated polymers and have prompted intensive studies of structure-function relationships in organic electronics[2,14–16,20,21]. In particular, taking

advantage of the widespread application of palladium-catalyzed polycondensations (including Stille coupling or Suzuki-Miyaura coupling)[1,22,23], much efforts have been devoted to developing DP-insensitive polymer donor ($P_D$) and polymer acceptor ($P_A$) materials in PSCs[24–26]. However, the aromaticity and reactivity of conjugated polymers undergo variations with the chain length and pre-aggregation among the living polymer chains during chain-growth polymerization. Consequently, solving the practical dilemma of polymer batch-to-batch (B2B) variations through molecular structure regulation is often insurmountable[14,24,25,27–29]. To this end, achieving high precision tailored DPs remains an important topic of high interest for investigating material properties and improving device performances.

Apart from material design[24,25], efforts to address the B2B variations in conjugated polymers have mainly focused on the development

[1]The Institute for Advanced Studies, Wuhan University, Wuhan 430072, China. [2]These authors contributed equally: Lin-Yong Xu, Wei Wang.
✉e-mail: sun.rui@whu.edu.cn; min.jie@whu.edu.cn

of suitable synthetic methods and process technologies capable of synthesizing polymer materials in a reproducible manner with small DP deviations[1,12,13,30–33]. Some Stille polycondensation reaction protocols, e.g., through examining the role of the solvent, catalyst, and other reaction conditions (e.g., microwave irradiation[31]) were found to improve the molecular mass ($M$s) and yields of conjugated polymers and decrease their polydispersity index (Đ) values[2,34,35]. For instance, using a stepwise-heating protocol in the Stille polycondensation in conjunction with optimized processing, Yang et al. obtained a high-performance representative conjugated $P_D$ PTB7 having a high weight-average molecular weight ($M_w$, 223 kDa), yield (85%) and narrow Đ (1.21)[1]. Moreover, Hwang and his coworkers investigated and synthesized PTB7 in a short reaction time with a specific molecular mass through a rapid-flow synthesis system[36]. Smeets et al. adopted a combination of Buchwald catalyst and droplet-flow chemistry to minimize B2B variations and structural defects of high-efficiency polymer donors PM6 and D18[37]. Despite this, recent synthesis experience tells us that even under the same reaction conditions and process protocols, there is usually no guarantee that the $M_w$ and Đ values of the conjugated polymers obtained each time are comparable. It is because, unlike other polymerization approaches such as ring-opening polymerization, palladium-catalyzed polycondensations with step-growth mechanisms are very sensitive to reaction conditions (including absolute humidity, reaction temperature and time, catalyst loading and purity, monomer purity, etc.)[36], generally leading to the fact that repeated trial and error processes on polymer synthesis and batch preparation[1,13–15,21]. Therefore, the currently reported attempts have not fundamentally surmounted the aforementioned B2B problems. This dilemma severely limits the development of polymer materials and the possibility of their commercial applications.

Recently, real-time, in-situ spectroscopy technologies have been valuable for developing and controlling polymerization reactions and detecting the DPs in real time[38–41]. For instance, both in-situ Fourier transform infrared and Raman spectroscopy have proven particularly useful to provide insight into key kinetic and material scalability[41], while eliminating the difficulties associated with offline measurements of polymerization reactions and their DPs, removing the disturbance of polymerization reaction (eg. the introduction of air and moisture)[39,41]. However, these real-time spectroscopy technologies can be used to monitor related addition polymerization (e.g., ring-opening polymerization) and condensation polymerization reactions[38–41], but not for quantitative analysis of Stille cross-coupling polymerization. Because the relevant characteristic peaks are difficult to identify, which will be further discussed below. Impressively, absorption and photoluminescence (PL) spectroscopy technologies can effectively distinguish the difference in spectral properties of π-conjugated donor-acceptor (D-A) conjugated polymers with various DPs[14,29,42,43]. This illustrates that both in-situ absorption and PL spectroscopy can provide possible pathways to obtain specific DP of a conjugated polymer in an offline manner during the polymerization reaction. However, in-situ PL technology possesses the irreplaceable advantages of simplicity, reliability, and sensitivity over absorption, FTIR, and Raman spectroscopy for organic semiconductor materials. This is because the absorptivity and fluorescence intensity of $P_A$s are usually positively correlated with the degree of conjugation over a range of molecular weights. When the molecular weight reaches a certain level, skeleton rigidity and polymer chain entanglement inhibit the vibration and rotation of molecular skeleton and functional groups. In this case, the emission spectra monitored by PL technology can dynamically respond to changes in the degree of radiative and non-radiative quenching of exciton online, providing a sensitive reflection of the physical and chemical state of the polymer chain compared to the other spectral techniques such as absorption, FTIR and Raman spectroscopy. It is worth noting that there are currently no in-situ spectroscopic technologies for real-time observations of Stille polycondensation reactions. Thus, it is important for the development and application of polymer materials to realize real-time monitoring of molecular mass during polymerization and to obtain functional conjugated polymers with desired DP by using in-situ PL spectroscopy technologies.

In this work, we develop an in-situ PL spectroscopy in tandem with a set of dedicated analysis and processing programs to monitor the oligomerization degree of the active materials during the polymerization reaction. The key feature of this tracking system is the ability to continue the high-quality iterative synthesis and preparation of already developed oligomers of PSCs while ensuring that there is no B2B variation in their device performances. We select the commercially available narrow-bandgap poly[(2,2′-((2Z,2′Z)-((12,13-bis(2-octyldodecyl)-3,9-diundecyl-12,13-dihydro[1,2,5]thiadiazolo[3,4e]thieno[2″,3″:4′,5′]-thieno[2′,3′:4,5]pyrrolo[3,2-g]thieno[2′,3′:4,5]thieno[3,2-b]-indole-2,10-diyl)bis(methanylylidene))bis(3-oxo-2,3-dihydro-1H-indene-2,1-diylidene))dimalononitrile-alt-2,5-thiophene)] (PYT)[14], developed by our laboratory as a test-bed polymeric material (Fig. 1a). We first present the influencing factors of PYT polymerization and the effects of DPs (or $M$s) on device performances. Importantly, we exhibit the correlation between the DP of PYT and its PL spectral features (including peak position ($PP$), peak intensity ($PI$), and peak position at the center of full width at half maximum () ($PPC$)). Based on these three parameters, we develop a real-time tracking protocol of the oligomerization degree of PYT in the Stille cross-coupling polymerization. This protocol can be used as an indication to determine the DP trends of oligomers during their polymerization reactions, thereby eliminating the time delays of sample extraction and offline analyses, and addressing the impact of B2B variations on device performance. In particular, we demonstrate that the reaction times for ideal PYT batches leading to the optimized performance can be controlled precisely with this protocol using different reaction conditions. We further investigate the applicability of PL monitoring techniques to Y-series $P_A$s with stereoregularity (PY-IT[44] and PY-OT[44]), Y-series $P_A$ with fluoro-substitution (PYF-T-$o$[45]), benzo[1,2-b:4,5-b′] dithiophene (BDT)-based $P_A$ (PTIB[46]) and $P_D$ (PM6[47]). The tracking system is readily adaptable to a wide range of general Y-series $P_A$s requiring high quality and desired DPs, which is broadly beneficial to the field of conjugated oligomer materials.

## Results

### A unified description of the dilemma of PYT polymerization

Narrow bandgap Y-series based $P_A$s are widely developed and used in all-polymer solar cells (all-PSCs) and have relatively high PCEs and low $M_w$s in the range of 10–50 kDa (approximately 4–20 repeat Y-series monomers) as shown in Supplementary Fig. 1 (extracted from Supplementary Table 1). Meanwhile, the spectrogram features of $P_A$s with various $M_w$s have obvious differences in solutions[14,48–51], which are very suitable for signal tracking and comparative analysis. As exhibited in Fig. 1a, herein we introduced a well-known $P_A$ PYT[14] as a test-bed polymer for tracking the DP evolution taking place during the Stille-polycondensation reactions. Considering the $M_w$ and yield of the polymer, we thus adopted Pd(PPh₃)₄ as the optimal catalyst to examine the following polymerization processes. Note that a simplified mechanism for Stille coupling is shown in Fig. 1b. The actual mechanism of Stille polymerization may vary according to different reaction conditions, including catalyst, ligands, solvent, monomer purity, etc[2,36].

It should be pointed out that we previously obtained three PYT batches with different $M_w$s (Supplementary Table 2)[14]. Furthermore, we used three polymer donors (see Supplementary Fig. 2 and Supplementary Table 2 for the molecular structures and molecular mass parameters of PBDB-T, PM6, and PM7, respectively) in combination with different PYT batches (named $PYT_L$ with a low $M_w$ of 9.7 kDa, $PYT_M$ with a medium $M_w$ of 10.6 kDa, and $PYT_H$ with a high $M_w$ of 16.1 kDa) to explore the relationship between device performance and PYT batches

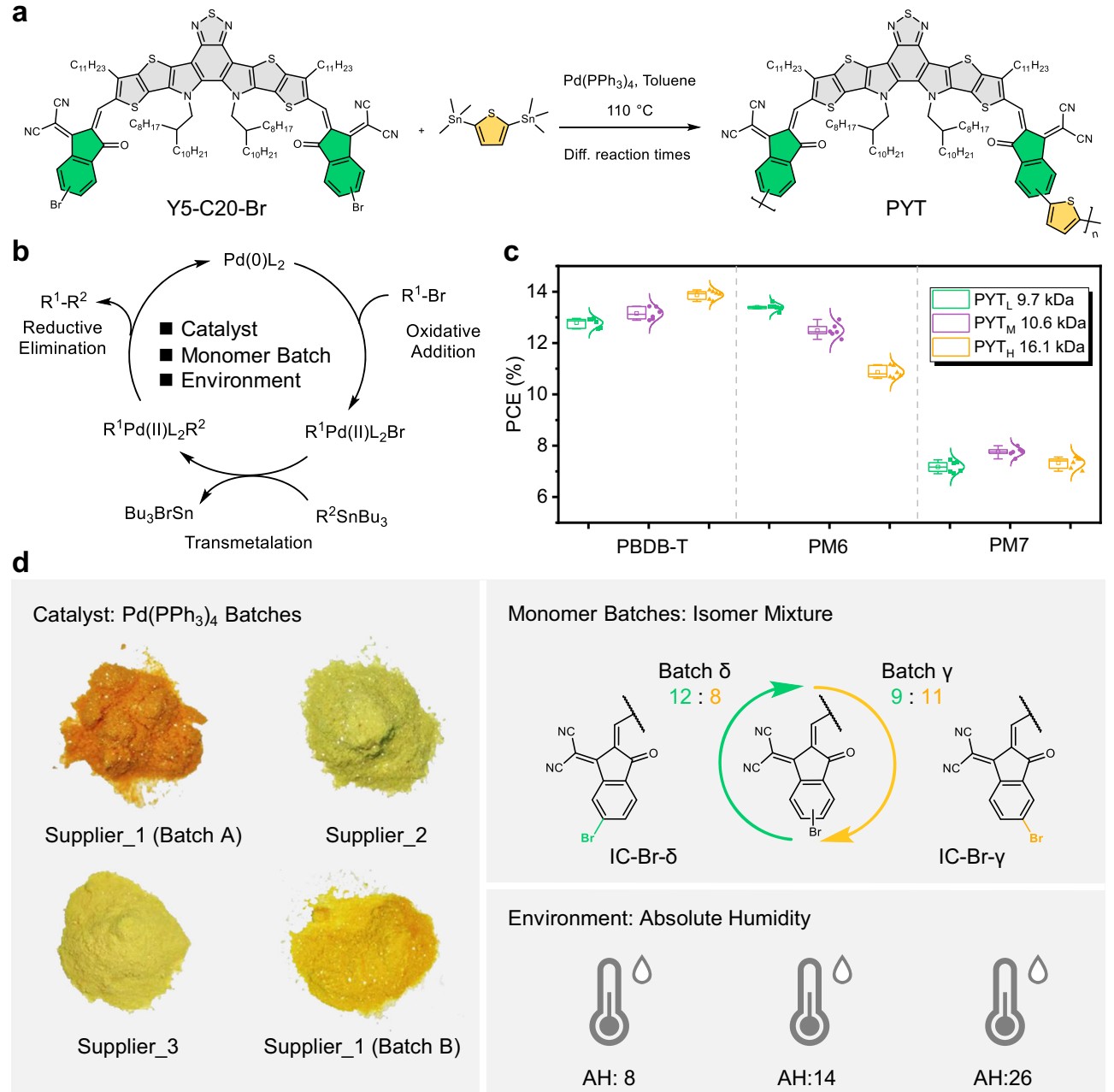

**Fig. 1 | Synthetic route of $P_A$ PYT and its description of the synthesis dilemma.** **a** Conventional Stille reaction of PYT comprised of only a steady heating procedure at 110 °C for different reaction times (50 min, 150 min, and 240 min). **b** The schematic of the transmetalation mechanism in Stille cross-coupling polymerization for the synthesis of PYT. **c** Photovoltaic performance characteristics of all-PSCs fabricated from PYT with PBDB-T, PM6, and PM7 based on different $M_w$s. **d** The influencing factors during polymerization: various suppliers of Pd(PPh$_3$)$_4$, monomer batch purification caused by isomers, and relevant environmental conditions.

(Supplementary Fig. 3 and Supplementary Table 3)[27]. As presented in Fig. 1c, a 3 × 3 all-PSC characterization/performance optimization matrix extracted from Supplementary Table 3 reveals dramatic variations in PCE ranging from 7.45% (PM7:PYT$_L$) to 14.07% (PBDB-T:PYT$_H$), mainly resulting from the differences in molecular miscibility and compatibility[27]. Further analysis is out of the scope of the present work. Despite this, we can easily find that PBDB-T matches PYT$_H$ better as compared to PYT$_L$ and PYT$_M$, on the contrary, PYT$_L$ and PYT$_M$ batches are more suitable for combination with PM6 and PM7, respectively (Fig. 1c). It's worth noting that the abovementioned results strongly reflect the current dilemma of polymer design and synthesis and the importance of developing real-time detection of the oligomerization degree[14,27,52].

Additionally, apart from the abovementioned reaction conditions (e.g., solvent, concentration, and reaction temperature), what we need to understand is that many other influencing factors, such as catalyst purity, monomer batches, environmental conditions, and so on, also significantly affect the rate and activity of polymerization, making it impossible to evaluate the oligomerization degree by reaction time[14]. For instance, we found that tetrakis(triphenylphosphine)-palladium(0) (Pd(PPh$_3$)$_4$) batches provided by three suppliers showed different properties, probably resulting from the oxidation degree of palladium(0) catalyst (Fig. 1d)[2,36]. Note that the different commercial products used in this study are denoted as Supplier_1 (Batches A and B), Supplier_2, and Supplier_3, respectively. We also noticed that the different molar ratios of δ/γ-brominated 1,1-dicyanomethylene-

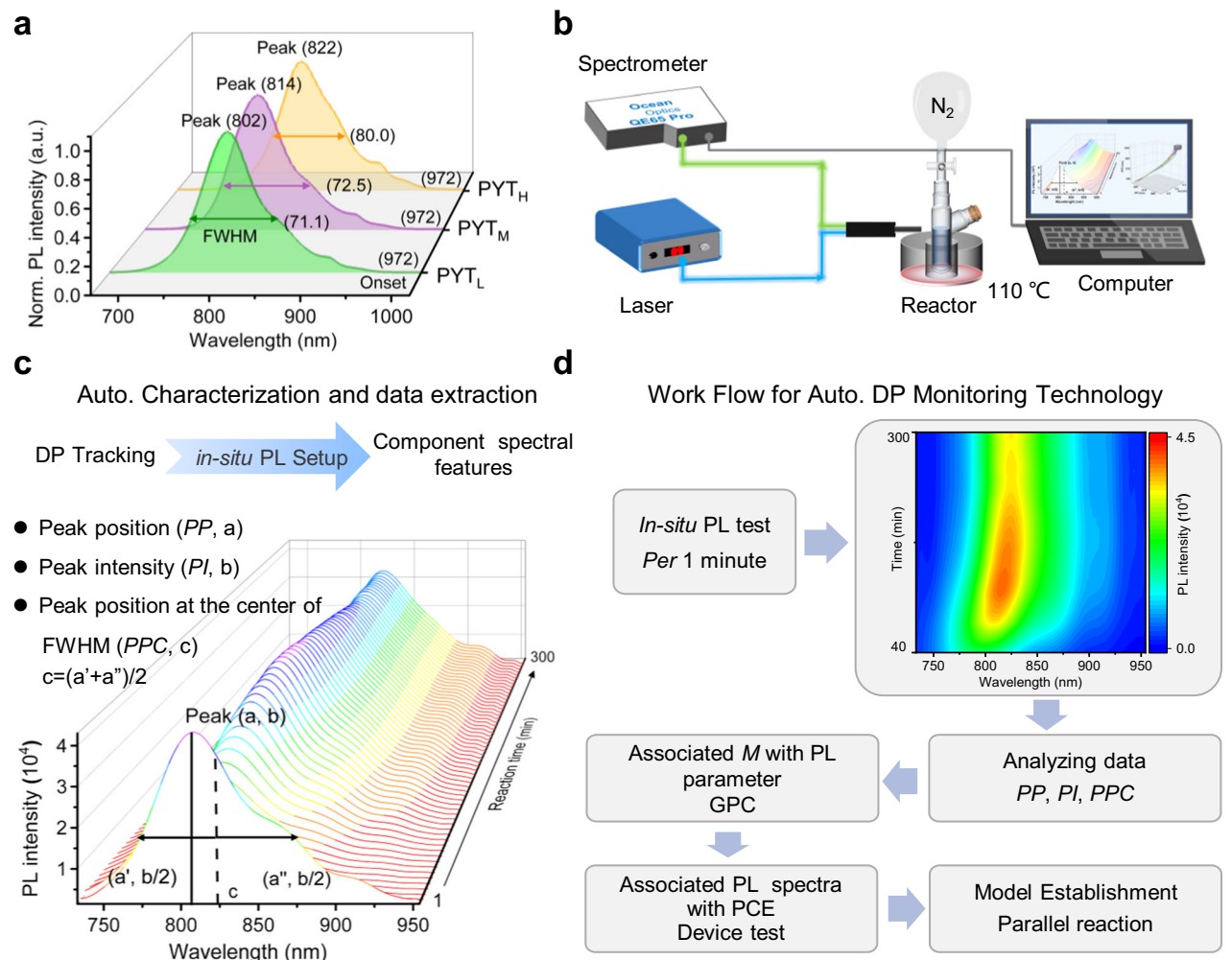

**Fig. 2 | Schematic explanation of the platform and workflow. a** Normalized PL spectra of the three PYT batches in chloroform solutions. **b** Schematic diagram of in-situ PL system for the synthesis of conjugated polymers and their DP detection. **c** Time traces of the PL spectra during the PYT polymerization reactions and relevant spectral parameters for DP tracking. The spectra were deconvoluted into different components that were quantified by *PP*, *PI*, and *PPC*. **d** Workflow for evaluating polymer synthesis with DP analysis in terms of relevant spectral parameters.

3-indanone isomeric end groups (IC-Br-δ and IC-Br-γ) in the isomerize mixture (Supplementary Fig. 4), which have different cross-coupling sites. This can also lead to differences in reaction rates and activities of PYT polymerization. Besides, the environmental conditions (Fig. 1d), the initial temperature setting, and the gradient all have an impact on the PYT polymerization. In short, the above experimental results and detailed discussion underscore the importance of $M_w$ control methodologies, especially Y-series-based $P_A$s (e.g., PYT as discussed above), that need to be improved and innovated to render all-PSCs commercially feasible.

**Setup design and workflow for DP monitoring**
Apart from the abovementioned effects of $M_w$s of PYT on device efficiencies, the strong influence of DP on the material properties relates to the absorption, IR, and PL spectra[14,29,52]. As an example, Supplementary Fig. 5 shows the measured absorption coefficients of the corresponding PYT batches in chloroform-diluted solutions with the same concentration. As compared to $PYT_L$ and $PYT_M$, $PYT_H$ exhibited a higher absorption coefficient, originating from the low π−π* excitation energy and high oscillator strength[19]. Owing to the high solution concentration of oligomers as well as the magnetic stirring process during PYT polymerization, it is not feasible to evaluate the oligomerization degree by real-time monitoring of absorption spectra

(Supplementary Fig. 6). In addition, we found that the IR spectra of the three PYT batches in solutions have no obvious signal differences (Supplementary Fig. 7), and the real-time characteristic peaks of the infrared spectra of the crude polymer production in the reaction flask are the same (Supplementary Fig. 8, obtained by a combined system composed of in-situ IR and in-situ PL modules (Supplementary Fig. 9)). Impressively, the observed DP dependence of spectral properties was exemplified by comparing PL data in PYT diluted solutions, as presented in Fig. 2a. Along with $M_w$ (or DP) increasing, the peak position and full width at half maximum (FWHM) of the related PYT batches increased gradually, even though their spectral onset was almost the same in solutions. These PL characteristics ensure that it is an excellent example to study the evolution of DP during the polymerization process.

The investigated spectral properties strongly indicate a substantial dependence on the $M_w$s, which provides an opportunity for an in-situ PL system to track DP in real time because it is an elegant and contactless method. Thus, we constructed an in-situ PL system for the synthesis of Y-series conjugated polymers, as depicted in Fig. 2b. In this combined system, apart from time traces of the PL spectra, relevant spectral parameters (including *PP*, *PI*, and *PPC*) can be extracted and analyzed in real-time, as illustrated in Fig. 2c. We simulate these PL spectra by using a parameterization. Building on the constructed

in-situ PL setup, we designed a detailed overview of the used analysis and processing programs, as presented in Supplementary Note 1 and Supplementary Code 1. The Python script can be scheduled to run every minute, depending on the specific settings. Moreover, this mathematical procedure allows us to extract and visualize the spectral parameters (including $PP$, $PI$, and $PPC$) of hundreds of PL spectra, and further evaluate the oligomerization degree during the poly-condensation reactions according to the evolution trends of these extracted parameters. Note that a Python script runs video based on the PYT synthesis is shown in Supplementary Movie 1, which does offer excellent online quality control.

The whole workflow for the automatic DP-monitoring technology is shown in Fig. 2d. In this workflow, to establish an effective model for a specific polymer, the products of multiple polymerizations were separately collected in the collecting chamber filled with methanol and further purified by methanol, acetone, and $n$-hexane. Subsequently, the trichloromethane fraction was concentrated to yield the resulting polymer. To minimize the interference of molecular weight by the purification operation, dichloromethane was not added as an eluent to further narrow the $Đ$ of the polymers, which may result in lower per-formance than reported in the literature. Furthermore, the $M_w$ and $Đ$ values for the resulting polymers are determined from high-temperature gel permeation chromatography (HT-GPC) measure-ments. Subsequently, the dataset from device fabrication and perfor-mance characterization was linked to the GPC and PL results for an overall evaluation in terms of DP and PCE for specific polymer synth-esis. Finally, relevant parallel experiments are carried out to verify the feasibility of the model. Note that Stille polycondensation contains many influencing factors, from the variables of which, in this work, four factors are selected by parallel experiments of the PYT synthesis to confirm the feasibility of the in-situ PL system and its prediction model, which will be discussed below.

## Data analysis for model building

To understand the relationships between $M_w$ and PL spectral char-acteristics (Fig. 2a) as well as any special correlations that may exist in the data, we used the designed in-situ PL grogram to monitor twelve PYT synthesis reactions as a function of different polymerization times. The total solution concentration, flask size, solvent, monomer and solvent loadings, environmental (an absolute humidity (AH) of $8-26 \, \mathrm{g \, m^{-3}}$) and internal (filled with nitrogen and not air) conditions, and the distance between reaction flask and laser, as well as other PL program settings, were fixed. The used Pd(PPh$_3$)$_4$ catalyst ensures that it comes from one batch, supplied by Supplier_1 (Batch A, Fig. 1d). Additionally, to minimize the possibility of stoichiometric imbalances from the impurity and B2B issue of the reactant monomers, we purified the D and A monomers. Since monomer A is an isomeric mixture of Y5-C20-Br-δ, Y5-C20-Br-γ, and Y5-C20-Br-δγ (Supplementary Fig. 2), we also specify the use of Y5-C20-Br from the same batch (Batch δ for IC-Br-δ:IC-Br-γ with a molar ratio of 12:8, Supplementary Fig. 4a) used in this PYT polymerization. Consequently, these Stille polycondensation conditions in terms of the reaction times allowed us to obtain PYT with different DPs (Supplementary Fig. 9).

The detailed variables extracted from Supplementary Fig. 10 are shown in Fig. 3a for $PP$, Fig. 3b for $PI$, and Fig. 3c for $PPC$, respectively. The $PP$ of the 12 sets of spectral data all showed a similar trend (Fig. 3d), as did spectral parameters $PI$ and $PPC$ (Fig. 3e, f). Note that the FWHM curves of the PL spectrum in these twelve reaction processes are not definite (Supplementary Fig. 11), and therefore not considered as a spectral parameter. The three spectral parameters ($PP$, $PI$, and $PPC$) shifted from 807 to 789 nm for $PP$, 817 to 794 nm for $PPC$, and $4.06 \times 10^4$ to $1.34 \times 10^4$ for $PI$ in the first 25 min, and then continuously increased to 824 nm for $PP$, 832 nm for $PPC$ in the next 200 min of the PYT polymerization reaction (Fig. 3d, f), and $3.96 \times 10^4$ for $PI$ in the next 100 min (Fig. 3e), respectively. Afterward, the $PP$ and $PPC$ increased

slowly between 225 and 300 min to reach their maximum values, then stabilized. The $PI$ in contrast decreased slowly between 125 and 300 min to reach a stable value of $3.40 \times 10^4$. To demonstrate the reproducibility of our system, the polymerization was repeated three times independently under the same reaction conditions. As shown in Supplementary Fig. 12, the two-dimensional PL images for each run are comparable.

We evaluated the $M_w$, number average molecular weight ($M_n$), and $Đ$ values of the twelve PYT batches by HT-GPC tests, as summarized in Supplementary Table 4, extracted from the GPC profiles (Supple-mentary Figs. 13 and 3g). Furthermore, the GPC parameters as a function of reaction time are shown in Fig. 3h. As expected, the $M_w$, $M_n$, and $Đ$ values gradually increased with increasing reaction time. Con-sequently, the $M_w$s of various PYT batches can be related to the cor-responding spectral parameters at the time of stopping the polymerization reactions (Fig. 3i). Based on this analysis, we can use the designed mathematical procedure to initially study the oligomer-ization degree of PYT during the real polymerization conditions. Additionally, after the fabrication and characterization experiments of the PBDB-T-, PM6- and PM7-based devices, the relationships between the device efficiencies and the $M_w$s of PYT batches can be performed (Fig. 3j), which are consistent with the above-discussed (Fig. 1c). Relevant photovoltaic parameters of the three all-polymer systems based on various PYT batches are summarized in Supplementary Fig. 14 and Supplementary Tables 5–7. To maximize the PCE of the PYT-based devices, we need to prepare $P_A$ PYT batches with higher $M_w$s for $P_D$ PBDB-T. On the contrary, to guarantee the higher PCE of the PM6:PYT system, we need to properly synthesize PYT with lower $M_w$ relative to the PBDB-T:PYT and PM7:PYT systems.

In particular, a representative time trace of the spectral para-meters ($PP$, $PI$, and $PPC$) detected from a PYT polymerization reaction is depicted in Fig. 3k. Through this three-dimensional graph, combined with the performance trends of the corresponding devices (Fig. 3j), we can achieve real-time monitoring of the oligomerization degree of PYT, and this graph enables a priori design capability (see Supple-mentary Movie 1) to eliminate B2B variations in the Stille poly-condensation. In short, these fundamental studies led us to establish the guiding principles for the design and operation of a real-time tracking DP system.

## Validation of the DP mathematical model

With the DP monitoring design rules of PYT established, we sought to demonstrate the ability of the developed in-situ PL tracking poly-merization reaction system in tandem with the mathematical proce-dure (Supplementary Note 1 and Supplementary Code 1) to produce precise qualities of PYT polymers over a wide range of different polymerization influencing factors. To determine the optimal combi-nation settings (Fig. 2b) for an effective DP monitoring of PYT, we examined the overall synthesis scope and fixed the reaction conditions with respect to parameter control such as solution concentration, catalyst loadings, solvent, and reaction temperature (Fig. 4a). Mean-while, we screened four important variables, including catalyst sup-pliers, oxidation time catalyst exposed to air, monomer batches (or purity), and ambient humidity (Fig. 4a). We only modulated one vari-able in the survey of the abovementioned four variables, and the other three variables were fixed, as shown in Table 1. Notably, since we introduced $P_D$ PBDB-T to fabricate the high-performance PYT-based all-polymer systems, the $M_w$ of the desired PYT batch is approximately 14.0 kDa, and the corresponding spectral parameters are 821 nm for $PP$, 831 nm for $PPC$ and $3.90 \times 10^4$ for $PI$, respectively. Thus, we can know when to stop the reaction to obtain the desired PYT batch by real-time tracking these three specific values during the polymeriza-tion process, as exhibited in Supplementary Movie 1.

We first selected four Pd(PPh$_3$)$_4$ catalyst batches from three cat-alyst suppliers. Intuitively from the color of the Pd(PPh$_3$)$_4$ catalysts

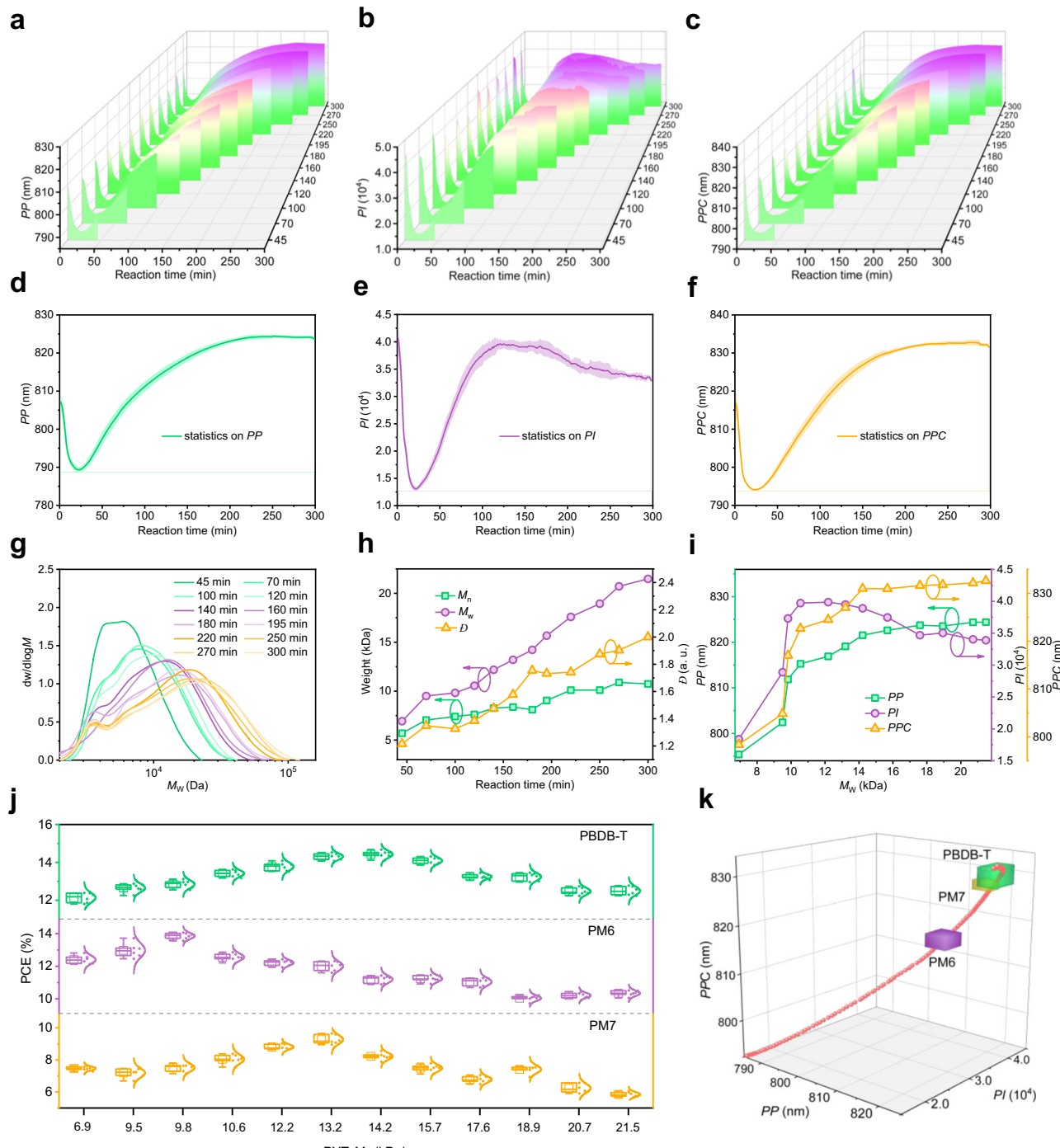

**Fig. 3 | Comparative analysis of online and offline experimental results.** The three-dimensional tendencies of (**a**) *PP*, (**b**) *PI*, and (**c**) *PPC* as a function of polymerization time, and the corresponding statistical graphs of spectral parameters (**d**) for *PP*, (**e**) for *PI*, and (**f**) for *PPC*. **g** GPC curves of relevant PYT batches as a function of polymerization time. **h** The tendency of $M_w$ variation as a function of polymerization time. **i** The spectral parameters of different PYT batches as a function of $M_w$. **j** The PCE *versus* $M_w$s of the PYT batches were investigated in the three all-polymer systems based on various $P_D$s. **k** Trends of the three parameters during the PYT polymerization, and the corresponding DP range (or related spectral parameters) of PYT that match well with PBDB-T, PM6, and PM7, respectively.

(Fig. 1d), they probably possess different material properties (like catalytic performance), the analysis of which is beyond the scope of this work. To achieve the spectral parameters set above (*PP* = 821 nm, *PPC* = 831 nm, and *PI* = 3.90 × 10⁴), the four PYT polymerization reaction times are 180 min for Supplier_1 (Batch A), 192 min for Supplier_2, 173 min for Supplier_3 and 186 min for Supplier_1 (Batch B), respectively. Although the reaction times of these four batches are different, the devices prepared by these four PYT batches all exhibited

comparable PCEs of approximately 14.5% (Supplementary Fig. 15 and Table 1), suggesting that the four PYT batches possess similar DPs supported by the GPC analysis (Supplementary Table 8, extracted from Supplementary Fig. 16). Additionally, taking batch A provided by Supplier_1 as an example, we further investigated the correlations between the reactivity of the palladium(0) catalyst and the reaction time of PYT polymerization through oxidizing the catalyst in the air, where catalysts of Batch_A0, Batch_A3, Batch_A7 and Batch_A15 are

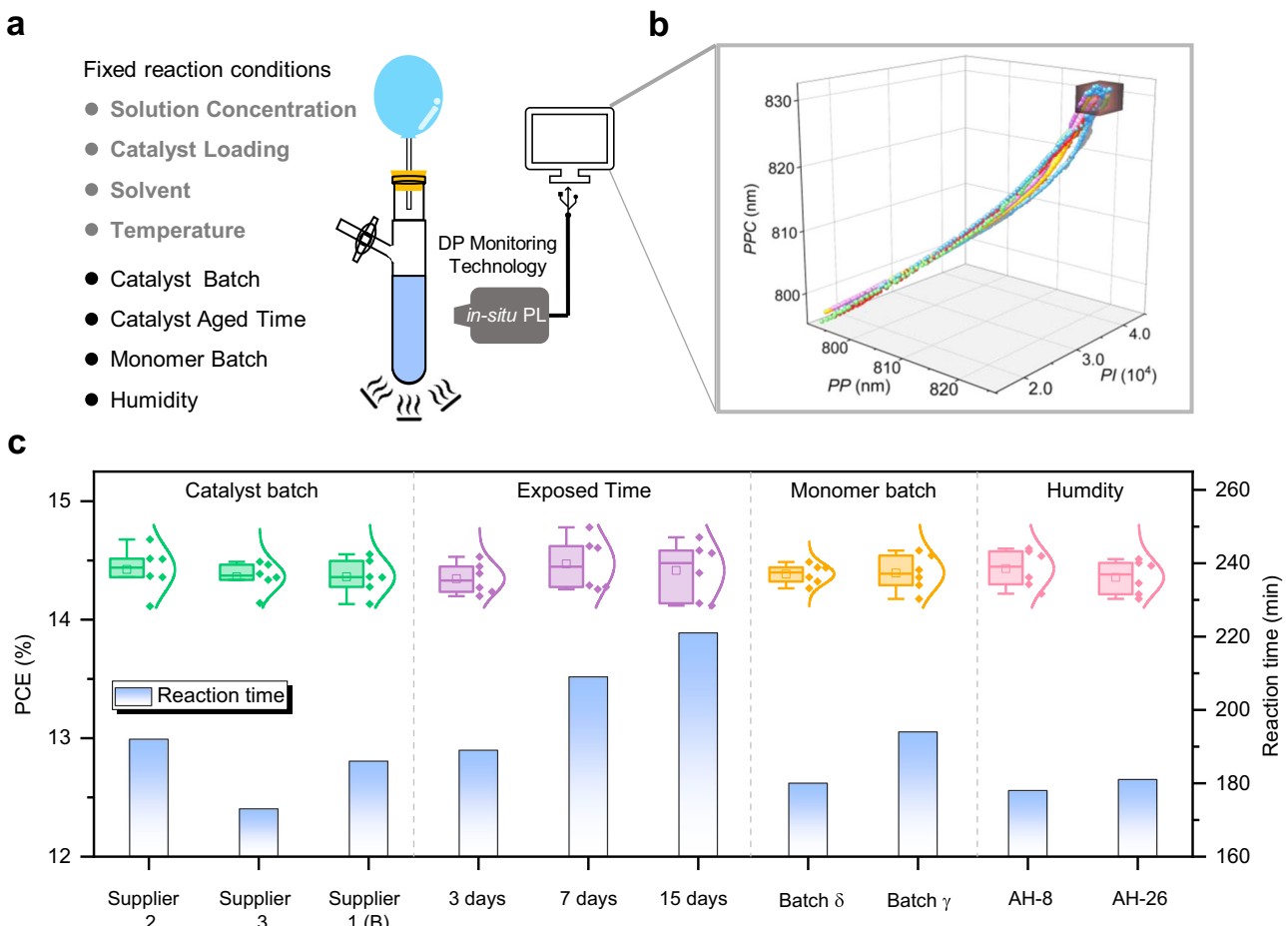

**Fig. 4 | Validation of oligomerization degree of PYT monitored by the protocol. a** Schematic illustration of the automated online characterization and relevant influencing factors in the polymerization reaction. **b** The traces of spectral parameters of ten PYT polymerization reactions based on various reaction conditions. Stop reacting when the parameter reaches the black box. **c** Polymerization time of PYT batches and their PCE values as a function of reaction condition.

exposed to air for 0, 3, 7 and 15 days, respectively (Supplementary Fig. 17). The reaction times of PYT polymerizations for achieving the specific spectral parameters are 180 min for Batch_A0, 189 min for Batch_A3, 209 min for Batch_A7, and 221 min for Batch_A15, respectively. As the degree of oxidation deepened, the reactivity of $Pd(PPh_3)_4$ catalyst gradually decreased, resulting in the prolongation of the reaction time to reach the desired DPs. Once again a qualitative match is observed with the targeted design, GPC, and PCE data. Relevant results from the $Pd(PPh_3)_4$ catalysts with four different oxidation times are summarized in Table 1. Besides, we found that the A-monomer batches based on Y5-C20-Br-δ:Y5-C20-Br-γ:Y5-C20-Br-δγ isomer mixtures can also influence the reaction times of PYT polymerization. As compared to the A-monomer batch based on the IC-Br-δ:IC-Br-γ molar ratio of 9:11 (194 min), the corresponding A-monomer batch composed of IC-Br-δ:IC-Br-γ isomers with a molar ratio of 12:8 exhibited a shorter reaction time of 180 min. Note that the influence of ambient humidity on reaction time was negligible (Table 1). We further analyzed the accuracy of the DP monitoring technology. As shown in Supplementary Fig. 18, the mean and standard deviation for the $M_w$ of the ten PYT batches are 14.45 and 0.45, respectively, with an accuracy of 97%, which achieved DP precise customization of PYT.

Figure 4b shows the traces of spectral parameters of ten PYT polymerization reactions based on various reaction conditions. The PYT reactions are terminated when the spectral parameters reach the set values, as depicted in Supplementary Movie 1. Notably, the rational adjustment of these four influencing factors was not consistent in obtaining reaction times for desired PYT batches (Table 1). Although

the reaction times are huge differences, the desired DPs can be obtained, which we determined by the same processing GPC analysis (Supplementary Table 8). To further confirm the accuracy of the correlation between the spectral parameters (or DPs of PYT) and device efficiencies, the performance results were analyzed by fabricating the PBDB-T:PYT devices based on ten PVT batches following the aforementioned optimized fabrication conditions, which are summarized in Fig. 4c. The average efficiencies of these all-polymer devices were all within the range of 14.49%–14.78% (Table 1 and Fig. 4c). These results demonstrate that our developed in-situ PL system in tandem with a set of dedicated analysis and processing programs is a great real-time detection technology for effectively monitoring the DP of the low-$M_w$ polymers, especially for PYT and its derivatives. The application of this technology can realize B2B preparation of high-performance functional polymers, which is conducive to industrial production and reducing costs.

## Universalization of the DP monitoring technology

Considering the effect of chemical structure modification on the spectra, we monitored the PL of regioregular (PY-IT[44] and PY-OT[44]), fluorinated (PYF-T-$o$[45]), BDT-based $P_{AS}$ (PTIB[46]) and $P_D$ (PM6[47]) under polymerization process (the chemical structures are listed in Fig. 5), to further explore the suitability and practicality of the DP monitoring technology. A new catalyst batch (Batch_C, Supplementary Fig. 19) was used for subsequent experiments. Similar to the process optimized for PYT, we controlled the reaction time to obtain polymers (batches A-E) for each system. The batch (X) with the optimal PCE then underwent interference

**Table 1 | Device parameters for PYT polymer batches as well as their polymerization reaction conditions and relevant reaction times**

| Batch | Reaction Time (min) | Catalyst | Exposed Time (day) | Monomer batch | AH (g m⁻³) | $V_{OC}$ (V) | $J_{SC}$ (mA cm⁻²) | FF (%) | PCEª (%) |
|---|---|---|---|---|---|---|---|---|---|
| Standard | 180 | Supplier_1 (Batch A) | 0 | δ | ~14 | 0.885 | 23.62 | 70.20 | 14.68 (14.35) |
| A | 192 | Supplier_2 | 0 | δ | ~14 | 0.902 | 23.21 | 69.21 | 14.49 (14.24) |
| B | 173 | Supplier_3 | 0 | δ | ~14 | 0.898 | 23.31 | 69.49 | 14.55 (14.32) |
| C | 186 | Supplier_1 (Batch B) | 0 | δ | ~14 | 0.891 | 23.57 | 69.20 | 14.53 (14.33) |
| D | 189 | Supplier_1 (Batch A) | 3 | δ | ~14 | 0.896 | 23.15 | 71.21 | 14.78 (14.49) |
| E | 209 | Supplier_1 (Batch A) | 7 | δ | ~14 | 0.891 | 23.77 | 69.33 | 14.69 (14.28) |
| F | 221 | Supplier_1 (Batch A) | 15 | δ | ~14 | 0.899 | 23.31 | 69.09 | 14.49 (14.30) |
| G | 194 | Supplier_1 (Batch A) | 0 | γ | ~14 | 0.892 | 23.70 | 68.98 | 14.58 (14.36) |
| H | 178 | Supplier_1 (Batch A) | 0 | δ | ~8 | 0.891 | 22.79 | 71.87 | 14.60 (14.35) |
| I | 181 | Supplier_1 (Batch A) | 0 | δ | ~26 | 0.887 | 24.02 | 68.08 | 14.51 (14.37) |

ªThe statistics were obtained from six devices.

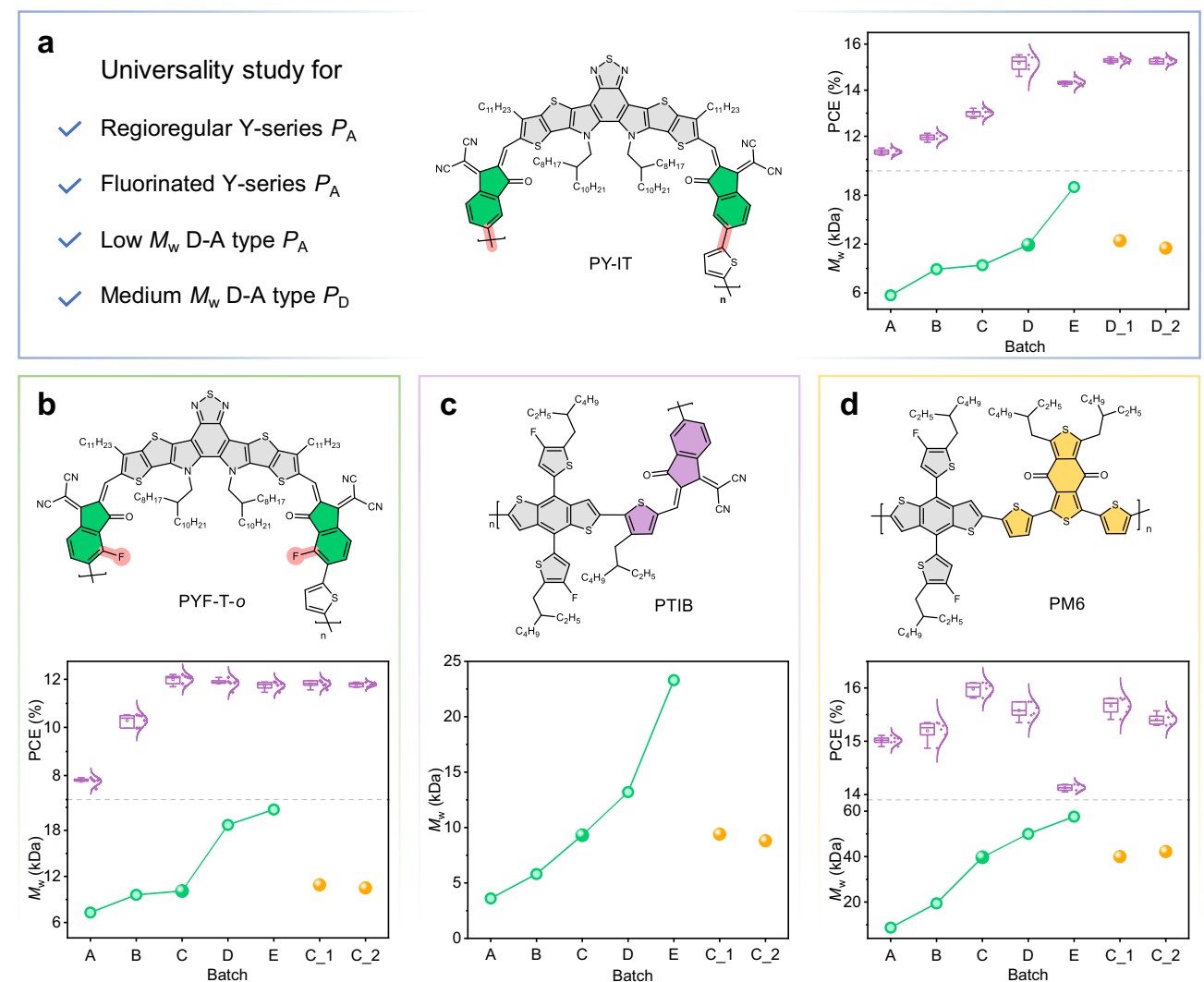

**Fig. 5 | The universality study results of the Y-series $P_A$s and BDT-based polymers.** The molecular structures of the investigated $P_A$ and $P_D$ materials. The $M_w$s and corresponding PCEs of the (**a**) PY-IT and (**b**) PYF-T-*o* batches were investigated in the PM6-based all-polymer systems. **c** The $M_w$s of the PTIB batches. **d** The $M_w$s and corresponding PCEs of PM6 batches investigated in the high-speed printing systems.

experiments to synthesize batches X_1 and X_2, varying the catalyst amount, which are 1.33 and 0.67 times than that of X respectively.

The 2D PL spectrum maps of PY-IT synthesized by polymerization of monomers Y5-C20-Br-γ and 2,5-bis(trimethylstannyl)thiophene are shown in Supplementary Fig. 20. Extracted *PP*, *PI* and *PPC* parameters versus time are presented in Supplementary Fig. 21, respectively. The three spectral parameters (*PP*, *PI*, and *PPC*) shifted from 796.6 to 786.3 nm for *PP*, 2.75 × 10⁴ to 1.41 × 10⁴ for *PI*, and 802.1 to 791.9 nm for

*PPC* in the first 69 min (PY-IT_A), and then, the three spectral parameters rapidly increased to (825.383 nm, $4.99 \times 10^4$ and 831.234 nm for PY-IT_B and PY-IT_C), followed by a slow increase to (830.316 nm, $4.91 \times 10^4$ and 834.717 nm for PY-IT_D). Afterward, these PL parameters reached stable as $M_w$ increased from 5.7 to 19.0 kDa across PY-IT batches A-E (Fig. 5a, Supplementary Fig. 22 and Supplementary Table 9). Device performance versus $M_w$ established for PY-IT showed batch D with the highest PCE of 15.53%. Batches D_1 and D_2 synthesized while varying catalyst, which showed the reaction time of 113 and 267 mins, respectively, almost replicated batch PY-IT_D $M_w$ (12.4 and 11.5 kDa) and PCE (15.44% and 15.42%) values (Supplementary Table 9, Supplementary Table 10 and Supplementary Fig. 23). The results demonstrate the realization of the B2B synthesis of PY-IT using the developed DP monitoring technique.

Additionally, we monitored the polymerization reaction of the PYT derivative PY-OT to better understand the effect of bromine isomerization at the terminal monomer groups. PY-OT was synthesized from monomers Y5-C20-Br-δ and 2,5-bis(trimethylstannyl)thiophene (Supplementary Fig. 2). As shown in Supplementary Fig. 24, the PL spectral parameters for PY-OT exhibited different trends compared to PY-IT. It can be found that the parameters began sharply decreasing after 100 min of reaction. Furthermore, some of the polymeric product precipitated out of solution when the reaction ceased, with an $M_w$ of 30.0 kDa. This observation suggests the bromine atom located closer to the cyano group in the monomer displays enhanced reactivity, likely due to the stronger electron-withdrawing properties of the cyano group rendering the carbon-bromine bond more susceptible to nucleophilic palladium-catalyzed insertion. In short, monitoring the polymerization reactions of PY-IT and PY-OT provided insight into how bromine isomerization at the monomer terminus can influence the reaction kinetics and polymer properties.

The effect of fluorine substitution on reactivity by monitoring the DP of a fluorinated PYT derivative, PYF-T-*o*, synthesized from monomers Y-OD-FBr-*o* (Supplementary Fig. 2) and 2,5-bis(trimethylstannyl) thiophene. The 2D PL spectrum maps of PYF-T-*o* polymerization process are listed in Supplementary Fig. 25, and corresponding spectral parameters are summarized in Supplementary Fig. 26. The $M_w$ for batches PYF-T-*o*_A-E ranged from 7.3 to 20.7 kDa (Fig. 5b, Supplementary Fig. 27 and Supplementary Table 11). The PCEs of relevant devices for these batches blended with PM6 ranged from 7.91% to 12.20% (Fig. 5b, Supplementary Fig. 28 and Supplementary Table 12). Batch PYF-T-*o*_ C was selected as a reference for verifying B2B synthesis. Note that batches PYF-T-*o*_C_1 and PYF-T-*o*_C_2 reached the set parameters ranges after 78 and 120 min, respectively, with $M_w$s of 10.9 and 10.5 kDa, respectively. The PCEs of both devices based on PYF-T-*o*_C_1 and PYF-T-*o*_C_2 are approximately 12.0%. Moreover, fluorine substitution activated the neighboring bromine, allowing an $M_w$ of 18.7 kDa after 170 min. However, it also enhanced aggregation-induced quenching compared to PY-IT, implying stronger F-S noncovalent interactions increase aggregation at high temperatures.

We extended the in-situ PL monitoring approach to non-Y-series systems to further assess its generalizability, exploring B2B synthesis of the BDT-based $P_A$ (PTIB[45], synthesized from BDT-TF-Sn and TIC-Br monomers (Supplementary Fig. 2)). The polymer possesses lower synthetic complexity and molecular weight than the reported high-performance D-A type $P_A$s, while demonstrating significant potential for scale-up fabrication. The 2D PL spectral maps, extracted parameters (*PP*, *PI*, *PPC*), and GPC data are shown in Supplementary Figs. 29–31 and Supplementary Table 13, respectively. Compared to Y-series polymers, PTIB exhibited weaker fluorescence at lower $M_w$ (<3.6 kDa, batch PTIB_A) and significant increases in PL parameters until $M_w$ reached ~9.3 kDa (batch PTIB_C). The parameters then slowly increased, stabilizing at (742.808 nm, $1.48 \times 10^4$, 756.406 nm) for batch PTIB_D with a $M_w$ of 13.2 kDa. In contrast, PP decreased for batch PTIB_E ($M_w$ > 13.2 kDa), possibly due to increased backbone distortion

for PTIB. Taking PTIB_C ($M_w$ ~ 9.3 kDa, reaction time = 183 min) as the reference, we further synthesized batches PTIB_C_1 and PTIB_C_2, which had $M_w$s of 9.4 and 8.8 kDa with the reaction time of 138 and 206 min, respectively (Fig. 5c).

Apart from the BDT-based $P_A$ PTIB, we also selected commercially available D-A polymer donor, PM6, to illustrate the advantages of this in-situ PL technique. The detailed trends of relevant parameters as well as the corresponding $M_w$s of different batches are demonstrated in Supplementary Figs. 32–34 and Supplementary Table 14, respectively. In accordance with our previous work[53], we conducted self-doping experiments using PM6 polymers of varying $M_w$s in the PM6:Y6 system. Our findings revealed that the medium $M_w$ variant (PM6_C) significantly enhances the high-speed processing of the active layer, while having minimal impact on its device performance. Thus, further taking batch PM6_C as an example, we synthesized other two batches PM6_C_1 and PM6_C_2, successfully attained the desired PL parameters range within 93 and 203 min, respectively. The corresponding $M_w$s were 39.0 kDa for PM6_C_1 and 42.2 kDa for PM6_C_2, with a precision range of ±1.90 kDa. As shown in Supplementary Fig. 35 and Supplementary Table 15, all three batches of doped materials (including PM6_C, PM6_C_1 and PM6_C_2, respectively) enabled the PM6:Y6 binary system to achieve PCEs exceeding 15.5% at a preparation speed of 30 m min$^{-1}$ in the air (Fig. 5d). The comparable device efficiencies demonstrated the successful achievement of B2B synthesis of PM6 additives suitable for high-throughput manufacturing. In summary, the validation studies demonstrate the general applicability of the in-situ PL monitoring technique in D-A conjugated polymers with low to medium $M_w$s.

## Discussion
In summary, we designed and developed an in-situ PL system in tandem with a set of dedicated analysis and processing procedures for monitoring the Stille polycondensation, applied it to the real-time DP detection of low-$M_w$ polymers, and successfully demonstrated our protocol for the PYT-based all-polymer systems. We first point out the B2B sensitivity of PYT on device performance and the fact that their polymerization reactions are generally influenced by various influencing factors, resulting in relatively poor DP control. Based on the investigations of the correlations between $M_w$ and PL spectra composed of specific spectral parameters (*PP*, *PI*, and *PPC*) as well as relevant device performances in three PYT-based all-polymer systems, spectral parameters, $M_w$ and PCE can strongly establish a well-defined association in a specific photovoltaic system. Using the developed online DP monitoring protocol and setting different polymerization reaction conditions, combined with the obtained correlation between the spectral parameters, the DP of PYT, and its PCE, ten quality controlled PYT batches can obtain a similar oligomerization degree and significantly overcome the B2B variations in device performance, leading excellent reproducibility in the overall PCEs of the PBDB-T:PYT system. The applicability of the technique was further verified in B2B synthesis of PY-IT, PYF-T-o, PTIB and PM6 polymers. Overall, our findings not only provide important progress for prevalent $P_A$ polymerization reactions but also help overcome the inveterate disadvantage of low-$M_w$ semiconducting polymerized under different polymer-based electronics applications, especially in terms of B2B variations of device performances. In particular, this real-time monitoring procedure is well suited for industrial scalability and desired material cost reduction.

## Methods
### Materials
PBDB-T, PM6 and PM7 were purchased from Solarmer Materials Inc. Toluene, BDT-TF-Sn, DT-BDD-Br and 2,5-bis(trimethylstannyl)thiophene for this work, were purchased from Chengdu Kelong Chemical Co., Ltd., SunaTech Inc. Toluene was dried and distilled from appropriate drying agents prior to use. Y5-C20-Br, Y5-C20-Br-γ, Y5-C20-Br-δ,

Y-OD-FBr-*o* and TIC-Br were synthesized according to literature and our previous work, and the corresponding NMR spectra are listed in Supplementary Figs. 4 and 36–40[14,44–46]. According to the [1]H NMR of monomer Y5-C20-Br, Batch δ and Batch γ show an isomer ratio of 12:8 and 9:11, respectively. The supplier of Supplier_1, Supplier_2 and Supplier_3 Pd(PPh₃)₄ catalysts are J&K Scientific Ltd., Energy Chemical (Sun Chemical Technology (Shanghai) Co., Ltd.) and Beijing Warwick Chemical Co., Ltd.

### Conventional stille polycondensation (Synthesis of PYT, PY-IT, PY-OT and PYF-T-*o*)

Y-series monomers (0.1 mmol) and 2,5-bis(trimethylstannyl)thiophene (0.1 mmol) were dissolved in 10 mL dry toluene in a customized reaction flask. After intensive bubbling with nitrogen for 10 min, 7.5 mg Pd(PPh₃)₄ was added to the reaction system. The mixture was kept in the preheated reactor at 110 °C for various durations, and the spectral data were recorded at intervals. After cooling to room temperature, the crude products were precipitated in methanol and further purified by Soxhlet extraction with methanol, acetone, hexane, and trichloromethane. The trichloromethane fraction was concentrated under reduced pressure and precipitated in methanol to obtain the resulting polymer.

### Conventional stille polycondensation (Synthesis of PTIB and PM6)

PTIB and PM6 were copolymerized from monomers TIB-Br and DT-BDD-Br, respectively, with monomer BDT-TF-Sn. The amount of Pd(PPh₃)₄ was 2.0 mg, and the other synthetic steps were consistent with Y-series $P_A$s.

### Materials characterization

[1]H NMR spectra were recorded on an AVANCE NEO 600 MHz spectrometer with *d*-chloroform as the solvent at room temperature. Absorption spectra were recorded with a Perkin-Elmer Lambda 365 UV-vis-NIR spectrophotometer from 300 nm to 1100 nm. The $M_n$, $M_w$ and $Đ$ values were quantitated by HT-GPC using 1,2,4-trichlorobenzene as the eluent at 160 °C and monodispersed polystyrene as the standard.

### General procedures for fabrication and characterization of the OSCs

All the devices were fabricated with a conventional structure architecture of Glass/ indium tin oxide (ITO)/ poly(3,4-ethylene dioxythiophene): poly(styrene sulfonate (PEDOT:PSS))/active layer/ cathode buffer layers/Ag. The cathode buffer layer of PYT-based devices is poly[9,9-bis[6(N, N, N-trimethylammonium)hexyl]fluorene-alt-co-1,4-phenylene]-bromide (PFN-Br), and the cathode buffer layer of other systems is PNDIT-F3N (9,9-bis(3′-(N,N-dimethylamino)propyl)−2,7-fluorene)-alt-5,5′-bis(2,2′-thiophene)−2,6-naphthalene-1,4,5,8-tetracaboxylic-N,N′-di(2-ethylhexyl)imide]. Pre-patterned ITO coated glass substrates (purchased from South China Science & Technology Company Limited) washed with methylbenzene, deionized water, acetone, and isopropyl alcohol in an ultrasonic bath for 15 min each. After blow-drying with high-purity nitrogen, all ITO substrates are cleaned in the ultraviolet ozone cleaning system (purchased from South China Science & Technology Company Limited) for 15 min. A thin layer of PEDOT:PSS (~40 nm) was deposited through spin-coating on pre-cleaned ITO-coated glass from a PEDOT:PSS aqueous solution (Xi'an Polymer Light Technology Corp 4083) at 4000 rpm for 30 s and dried subsequently at 150 °C for 15 min in atmospheric air. Then the active layer was spin-coated from CHCl₃ solution (1-chloronaphthalene as the additive) with the optimal concentration and donor/acceptor weight ratios, which were further annealed. The cathode buffer layers via a solution concentration of 0.5 mg mL⁻¹ were deposited on the top of the active layer at a rate of 4000 rpm for 30 s. Finally, the top silver

electrode of 100 nm thickness was thermally evaporated through a mask onto the cathode buffer layer under a vacuum of ~5 × 10⁻⁶ mbar. The optimal active layer thickness measured by a Bruker Dektak XT stylus profilometer was about 100 nm. The typical active area of the investigated devices was 5 mm².

The PBDB-T:PYT active layer was spin-coated from CHCl₃ solution with the optimal donor/acceptor weight ratios of 1:0.75 with 3 vol% of 1-CN (a total concentration of 17 mg mL⁻¹), which were further annealed at 100 °C for 10 min. The PM6:PYT and PM7:PYT active layers were spin-coated from CHCl₃ solution with the optimal donor/acceptor weight ratios of 1:1.5 with 1.5 vol% of 1-CN (a total concentration of 16 mg mL⁻¹), which were further annealed at 100 °C for 5 min. The PM6:PY-IT active layer was spin-coated from CHCl₃ solution with the optimal donor/acceptor weight ratios of 1:1 with 1 vol% of 1-CN (a total concentration of 14 mg mL⁻¹), which were further annealed at 95 °C for 5 min. The PM6:PYF-T-*o* active layer was spin-coated from CHCl₃ solution with the optimal donor/acceptor weight ratios of 1:1.2 with 2 vol% of 1-CN (a total concentration of 16 mg mL⁻¹), which were further annealed at 95 °C for 5 min. The PM6:Y6 active layer was blade-coated at a rate of 30 m min⁻¹ in the air from a solution of PM6:Y6 (1:1.2, with 5 wt% PM6 which was synthesized in this work) with 6.0 mg mL⁻¹ solution concentrations in chloroform.

The current-voltage characteristics of the solar cells were measured by a Keithley 2400 source meter unit under AM1.5 G (100 mW cm⁻²) irradiation from a solar simulator (Enlitech model SS-F5-3A). Solar simulator illumination intensity was determined at 100 mW cm⁻² using a monocrystalline silicon reference cell with KG5 filter. Short circuit currents under AM1.5 G (100 mW cm⁻²) conditions were estimated from the spectral response and convolution with the solar spectrum. The external quantum efficiency was measured by a Solar Cell Spectral Response Measurement System QE-R3011(Enlitech Technology Co., Ltd.).

### Photoluminescence measurements

The PL excitation wavelength was set to 405 nm using a Laser-405-1HS. PL data were collected using a QE65 Pro spectrometer.

### General procedure for in-situ PL system

Set up the in-situ PL device according to the schematic diagram (Fig. 2b), and the mixture was kept in the preheated reactor at 110 °C for various durations. During the process, the laser emits a laser beam of 405 nm every minute while the spectrometer receives PL spectrum and generates relevant data. The Python script can be scheduled to run and extract the three spectral parameters (*PP*, *PI*, and *PPC*), depending on the specific settings.

### Reporting summary

Further information on research design is available in the Nature Portfolio Reporting Summary linked to this article.

## Data availability

The authors declare that the source data generated in this study are provided in the Supplementary Information and Source Data file. All source data generated during the current study are available from the corresponding authors upon request. Source data are provided with this paper.

## Code availability

The computational codes used to generate results of this study are described in Supplementary Note 1 and Supplementary Code 1.

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

## Acknowledgements

J.M. is grateful for the financial support from the National Natural Science Foundation of China (NSFC) (Grant No. 52061135206 and 22279094) and the Fundamental Research Funds for the Central Universities.

## Author contributions

J.M., W.W. and L.-Y.X. conceived the ideas. L.-Y.X. and W.W. performed synthesis experiments. R.S. and S.S.W. fabricated all the solar cell samples. L.-Y.X. designed the analytical processing program. L.-Y.X., W.W., X.R.Y., Y.M.S. and M.X.C. performed data analysis. J.M., L.-Y.X. and W.W. contributed to manuscript preparation, and L.-Y.X. and W.W. supervised by J.M. conceived and directed the project. All authors commented on the manuscript.

## Competing interests

The authors declare no competing interests.
