## [Peer Review File · Nature Communications]

Real-time monitoring polymerization degree of organic photovoltaic materials toward no batch-to-batch variations in device performanceREVIEWER COMMENTS

Reviewer #1 (Remarks to the Author):

Xu and Wang et al. monitored the degree of polymerization using in-situ photoluminescence (PL) system to obtain polymers with low batch-to-batch variation and to yield reproducible performance of OSCs. Although authors claimed that in-situ PL has not been studied for real time monitoring of Stille cross-coupling polymerization, it is under the mainstream of spectroscopic techniques and still the same concept of monitoring absorption, FTIR, Raman etc. The spectroscopic techniques are very common to monitor the DP and have been widely utilized in several fields of synthesis. In addition, despite reproducible PCE demonstrated by PBDB-T:PYT system, this method may not guarantee for other polymer donor-acceptor system and further study is required to validate the claim. Hence, the manuscript is more suitable for specialized journals, rather than multidisciplinary journal like Nat. Comm.

Reviewer #2 (Remarks to the Author):

In this manuscript, Xu et al report the use of photoluminescence (PL) spectroscopy for in-situ monitoring of the degree of polymerization of polymer acceptors designed for organic photovoltaics (OPV) during its synthesis via Stille polymerization. For this polymer acceptor system (namely PYT), its performance in OPV devices has been previously found to depend strongly on its degree of polymerization/oligomerization, but currently there is a lack of method for real-time monitoring of the polymerization degree during its synthesis that leads to large batch-to-batch variations in device performance. The in-situ PL method reported in this manuscript tracks the change in (1) PL peak position, (2) FWHM of PL spectrum and (3) PL peak intensity as a function of time (minutes) during the polymerization process. Since these PL spectral parameters change with increasing polymerization degree, the authors were able to establish a quantitative model and achieve in-situ monitoring of polymerization over reaction time and controlled synthesis of PYT with varying molecular weights.

Overall this manuscript reports a practical method for improving the synthesis procedure of polymers useful for all-polymer OPV devices, which is important for future research and development of this promising technology. However, I feel this manuscript may be more suitable for a more specialized journal as the results do not contain significant scientific novelty. Some technical comments for consideration are listed below.

1) It may be helpful to discuss about the accuracy of this method in determining the material's molecular weight. How this accuracy may change vary between different types of polymers should also be discussed.

2) Fig 3G, H: units for time and weight are missing.

Reviewer #3 (Remarks to the Author):

The manuscript by Xu et al investigates the real-time monitoring of the oligomerization degree (OD) of polymer acceptors to acquire high-quality polymer acceptor (PYT) with no batch-to-batch (B2B) variations, which is benefit for fabricating high performance all-polymer solar cells. Herein, Xu et al develop an in-situ photoluminescence (PL) system to track and estimate the OD through the Stille polymerization. Moreover, they establish a mathematical model for the quantitative prediction of the experimental molecular weight and device efficiencies. As a result, the similar PCEs have been achieved through the monitor at different condition, which is promising and potential for the industrial scalability and desired cost reduction in the future. The publication of the manuscript in this prestigious journal seems to be appropriate if the questions and comments below are fully addressed.

1. The Mws of the three PYT batches are supposed to be provided in the article (Page 5, Line 135) and Figure 1c, which makes the comparison between different batches clearer.
2. The reaction temperature should be marked on the reaction arrow in Figure 1a. The name of Figure 1c seems to miss PM6 and PM7.
3. Please explain why the peak intensity (PP) of PYT increases firstly and then weakens.
4. Figure 3l seems to miss the arrows to verify the ownership of lines and vertical coordinate.
5. Please give detailed exposed time in the article because it appears in Figure 4c and Table 1.
6. Please carefully check if the titles of the Figures and Tables are correct (including capitalization, punctuation, grammar, etc.)
7. This work seems to lack universal validation. I wonder if this method is also suitable for the synthesis of other polymer acceptors (such as PAs with stereoregularity)?

Response to Referee #1:

Comments to the Author: *Xu and Wang et al. monitored the degree of polymerization using in-situ photoluminescence (PL) system to obtain polymers with low batch-to-batch variation and to yield reproducible performance of OSCs. Although authors claimed that in-situ PL has not been studied for real time monitoring of Stille cross-coupling polymerization, it is under the mainstream of spectroscopic techniques and still the same concept of monitoring absorption, FTIR, Raman etc. The spectroscopic techniques are very common to monitor the DP and have been widely utilized in several fields of synthesis. In addition, despite reproducible PCE demonstrated by PBDB-T:PYT system, this method may not guarantee for other polymer donor-acceptor system and further study is required to validate the claim. Hence, the manuscript is more suitable for specialized journals, rather than multidisciplinary journal like Nat. Comm.*

Response: We highly appreciate the reviewer's insightful comments on our manuscript. We have provided the necessary revisions and descriptions in the revised manuscript. In the following, we give a point-by-point reply to your comments.

Here we would like to further provide our viewpoints:

1. While other spectroscopic techniques are used to monitor polymerization (real-time, in-situ FTIR and Raman Spectroscopy provide enhanced knowledge and improved performance in the investigation of polymerization reactions (like, analyzing broad range of polymerization reactions, including homogeneous (e.g. free radical and condensation) and heterogeneous (e.g. emulsion and microemulsion))), the claim is that in-situ PL has not been studied for real-time monitoring of Stille cross-coupling polymerization, and real time, in-situ FTIR, Raman and Absorption Spectroscopy cannot be used to monitor this type of polymerization reaction. We demonstrate the application and potential advantages of in-situ PL technology through our work, which is much important to improve the polymerization reactions of donor-acceptor fundamental polymer materials for all kinds of organic electronics. Just because a

technique is new to a specific reaction does not diminish its significance.

2. In the revised manuscript, we showed this technique worked well for the other polymer donor and acceptor materials (including PY-OT, PYF-T-o, PM6 and PTIB) with suitable molecular weights, strongly demonstrating the approach's validity.
3. The manuscript presents a new analytical technique with potential interdisciplinary applications, not just specialized results. Journals like Nat. Commun. aim to advance science across disciplines, which this work does by introducing in-situ PL monitoring of polymerization. This work introduces an innovative monitoring methodology for an important application area. We believe the development of new analytical techniques can benefit multiple research and industrial communities.

In summary, the work's innovations and significance for cross-disciplinary science are obvious. We hope that introducing a new method for an important problem like polymerization reaction of conjugated polymers has wide-reaching implications worthy of a journal like Nat. Commun.

Comments:

Comment 1. *Although authors claimed that in-situ PL has not been studied for real time monitoring of Stille cross-coupling polymerization, it is under the mainstream of spectroscopic techniques and still the same concept of monitoring absorption, FTIR, Raman etc. The spectroscopic techniques are very common to monitor the DP and have been widely utilized in several fields of synthesis.*

Response: Thanks for the reviewer's comment. For organic semiconductor materials, *in-situ* PL technology has the irreplaceable advantages of simplicity, reliability, and sensitivity over absorption, FTIR, and Raman spectroscopy, which is one of the reasons why we adopted *in-situ* PL to monitor the molecular weight of P_{AS} . This is because the

absorptivity and fluorescence intensity of P_{AS} are usually positively correlated with the degree of conjugation over a range of molecular weights. When the molecular weight reaches a certain level, skeleton rigidity and polymer chain entanglement inhibit the vibration and rotation of molecular skeleton and functional groups. In this case, the emission spectra can dynamically respond to changes in the degree of radiative and non-radiative quenching of exciton (Aggregation-Induced Quenching, AIQ), providing a sensitive reflection of the physical and chemical state of the polymer chain compared to techniques such as absorption, FTIR and Raman spectroscopy. Therefore, it is important that we have achieved the first B2B synthesis of oligomers using *in-situ* PL monitoring technology.

It's worth noting that that other techniques such as absorption, FTIR and Raman spectroscopy can not be used to monitor the polymerization reactions of conjugated polymer materials: Absorption spectroscopy only provides information about the overall conjugation/delocalization in the polymer backbone, but cannot distinguish polymer chains of different lengths or end groups. Meanwhile, the concentration of the solution in the polymerization reaction is very large and usually exceeds the detection range of absorption. These make it difficult to accurately monitor molecular weight/degree of polymerization. FTIR is only sensitive to certain functional groups and vibrational modes. The spectra can become very complex for conjugated polymers, making it challenging to deconvolute and assign peaks specifically to the polymerization process. Raman spectroscopy suffers from strong fluorescence interference from conjugated polymers, limiting its ability to obtain high quality spectra *in situ* during the reaction. Of particular note is that we tried to use FTIR and absorption spectroscopy techniques to monitor the polymerization reactions of conjugated PYT polymers. However, we did not succeed, resulting from the abovementioned reasons. Besides, inhomogeneous reaction mixtures and insoluble polymer byproducts can complicate data analysis and interpretation of absorption/FTIR/Raman spectra. In contrast, PL spectroscopy provides a direct optical probe of the electronic structure, which are highly sensitive to polymer chain length/end groups. PL can more selectively

and quantitatively monitor the polymerization with high signal-to-noise, overcoming the limitations of other spectroscopic techniques for this application.

We also added the necessary descriptions in the revised main text, as follows:

“However, these real-time spectroscopy technologies can be used to monitor related addition polymerization (e.g. ring-opening polymerization) and condensation polymerization reactions, but not for quantitative analysis of Stille cross-coupling polymerization. Because the relevant characteristic peaks are difficult to identify, which will be further discussed below. Impressively, absorption and photoluminescence (PL) spectroscopy techniques can effectively distinguish the difference in spectral properties of π -conjugated donor-acceptor (D-A) conjugated polymers with various DPs. This illustrates that both in-situ absorption and PL spectroscopy can provide possible pathways to obtain specific DP of a conjugated polymer in an offline manner during the polymerization reaction. It is worth noting that there are currently no in-situ spectroscopic techniques for real-time observations of Stille polycondensation reactions.”;

“Owing to the high solution concentration of oligomers as well as the magnetic stirring process during PYT polymerization, it is not feasible to evaluate the oligomerization degree by real-time monitoring of absorption spectra. In addition, we found that the IR spectra of the three PYT batches in solutions have no obvious signal differences (Supplementary Fig. 6), and the real-time characteristic peaks of the infrared spectra of the crude polymer production in the reaction flask are the same (Supplementary Fig. 7, obtained by a combined system composed of in-situ IR and in-situ PL modules (Supplementary Fig. 8)).”.

Comment 2. *In addition, despite reproducible PCE demonstrated by PBDB-T:PYT system, this method may not guarantee for other polymer donor-acceptor system and further study is required to validate the claim.*

Response: Many thanks for the reviewer's comment and suggestion. In the revised manuscript, we further investigated the applicability of PL monitoring technology to Y-series P_{AS} with stereoregularity (PY-IT and PY-OT), Y-series P_A with fluoro-substitution (PYF-T-*o*), BDT-based P_A (PTIB) and BDT-based P_D (PM6). As shown in Figure 5 in the revised main text (extracted from Supplementary Figs. 19-34 and Supplementary Tables 9-15 in the revised SI file), the validation studies demonstrate the general applicability of the *in-situ* PL monitoring technique in D-A conjugated polymers with low to medium M_{ws} .

We added the corresponding descriptions in the section of “*Universalization of the DP monitoring technology*” in the revised main text, as follows:

“*Considering the effect of chemical structure modification on the spectra, we monitored the PL of regioregular (PY-IT⁴³ and PY-OT⁴³), fluorinated (PYF-T-*o*⁴⁴), BDT-based (PTIB⁴⁵) P_{AS} and BDT-based P_D (PM6⁴⁶) under polymerization process (the chemical structures are listed in **Fig. 5A**), to further explore the suitability and practicality of the DP monitoring technology. A new catalyst batch (Batch_C, **Supplemental Fig. 18**) was used for subsequent experiments. Similar to the process optimized for PYT, we controlled the reaction time to obtain polymers (batches A-E) for each system. The batch (X) with the optimal PCE then underwent interference experiments to synthesize batches X_1 and X_2, varying the catalyst amount, which are 1.33 and 0.67 times than that of X respectively.”;*

“*The 2D PL spectrum maps of PY-IT synthesized by polymerization of monomers Y5-C20-Br- γ and 2,5-bis(trimethylstannyl)thiophene are shown in **Supplemental Fig. 19**. Extracted PP, PI and PPC parameters versus time are presented in **Supplemental Fig. 20**, respectively. The three spectral parameters (PP, PI, and PPC) shifted from 796.6 to 786.3 nm for PP, 2.75×10^4 to 1.41×10^4 for PI, and 802.1 to 791.9 nm for PPC in the first 69 min (PY-IT_A), and then, the three spectral parameters rapidly increased to (825.383 nm, 4.99×10^4 and 831.234 nm for PY-IT_B and PY-IT_C), followed by a slow increase to (830.316 nm, 4.91×10^4 and 834.717 nm for PY-IT_D). Afterwards, these PL*

parameters reached stable as M_w increased from 5.7 to 19.0 kDa across PY-IT batches A-E (Fig. 5B, Supplemental Fig. 21 and Supplemental Table 9). Device performance versus M_w established for PY-IT showed batch D with the highest PCE of 15.53%. Batches D_1 and D_2 synthesized while varying catalyst, which showed the reaction time of 113 and 267 mins, respectively, almost replicated batch PY-IT_D's M_w (12.4 and 11.5 kDa) and PCE (15.44% and 15.42%) values (Supplemental Table 9, Supplemental Table 10 and Supplemental Fig. 22). The results demonstrate the realization of the B2B synthesis of PY-IT using the developed DP monitoring technique.”;

“Additionally, we monitored the polymerization reaction of the PYT derivative PY-OT to better understand the effect of bromine isomerization at the terminal monomer groups. PY-OT was synthesized from monomers Y5-C20-Br- δ and 2,5-bis(trimethylstannyl)thiophene (Supplemental Fig. 2). As shown in Supplemental Fig. 23, the PL spectral parameters for PY-OT exhibited different trends compared to PY-IT. It can be found that the parameters began sharply decreasing after 100 minutes of reaction. Furthermore, some of the polymeric product precipitated out of solution when the reaction ceased, with a M_w of 30.0 kDa. This observation suggests the bromine atom located closer to the cyano group in the monomer displays enhanced reactivity, likely due to the stronger electron-withdrawing properties of the cyano group rendering the carbon-bromine bond more susceptible to nucleophilic palladium-catalyzed insertion. In short, monitoring the polymerization reactions of PY-IT and PY-OT provided insight into how bromine isomerization at the monomer terminus can influence the reaction kinetics and polymer properties.”;

“The effect of fluorine substitution on reactivity by monitoring the DP of a fluorinated PYT derivative, PYF-T-o, synthesized from monomers Y-OD-FBr-o (Supplemental Fig. 2) and 2,5-bis(trimethylstannyl)thiophene. The 2D PL spectrum maps of PYF-T-o polymerization process are listed in Supplemental Fig. 24, and corresponding spectral parameters are summarized in Supplemental Fig. 25. The M_w for batches PYF-T-o_A-

E ranged from 7.3-20.7 kDa (**Fig. 5C, Supplemental Fig. 26 and Supplemental Table II**). The PCEs of relevant devices for these batches blended with PM6 ranged from 7.91%-12.20% (**Fig. 5C, Supplemental Fig. 27 and Supplemental Table 12**). Batch PYF-T-o_C was selected as a reference for verifying B2B synthesis. Note that batches PYF-T-o_C_1 and PYF-T-o_C_2 reached the set parameters ranges after 78 and 120 mins, respectively, with M_{ws} of 10.9 and 10.5 kDa, respectively. The PCEs of both devices based on PYF-T-o_C_1 and PYF-T-o_C_2 are approximately 12.0%. Moreover, fluorine substitution activated the neighboring bromine, allowing a M_n of 18.7 kDa after 170 minutes. However, it also enhanced aggregation-induced quenching compared to PY-IT, implying stronger F-S noncovalent interactions increase aggregation at high temperatures.”;

“We extended the in-situ PL monitoring approach to non-Y-series systems to further assess its generalizability, exploring B2B synthesis of the BDT-based P_A (PTIB⁴⁵, synthesized from BDT-TF-Sn and TIC-Br monomers (**Supplemental Fig. 2**)). The 2D PL spectral maps, extracted parameters (PP, PI, PPC), and GPC data are shown in **Supplementary Figs. 28-30 and Supplementary Table 13**, respectively. Compared to Y-series polymers, PTIB exhibited weaker fluorescence at lower M_w (<3.6 kDa, batch PTIB_A) and significant increases in PL parameters until M_w reached ~9.3 kDa (batch PTIB_C). The parameters then slowly increased, stabilizing at (742.808 nm, 1.48×10^4 , 756.406 nm) for batch PTIB_D with a M_w of 13.2 kDa. In contrast, PP decreased for batch PTIB_E ($M_w > 13.2$ kDa), possibly due to increased backbone distortion for PTIB. Taking PTIB_C ($M_w \sim 9.3$ kDa, reaction time = 183 mins) as the reference, we further synthesized batches PTIB_C_1 and PTIB_C_2, which had M_{ws} of 9.4 and 8.8 kDa with the reaction time of 138 and 206 mins, respectively.”;

“Apart from the BDT-based P_A PTIB, we also selected commercially available D-A polymer donor, PM6, to illustrate the advantages of this in-situ PL technique. The detailed trends of relevant parameters as well as the corresponding M_{ws} of different batches are demonstrated in **Supplementary Figs. 31-33 and Supplementary Table 14**,

respectively. In accordance with our previous work,⁵¹ we conducted self-doping experiments using PM6 polymers of varying M_w s in the PM6:Y6 system. Our findings revealed that the medium M_w variant (PM6_C) significantly enhances the high-speed processing of the active layer, while having minimal impact on its device performance. Thus, further taking batch PM6_C as an example, we synthesized other two batches PM6_C_1 and PM6_C_2, successfully attained the desired PL parameters range within 93 and 203 minutes, respectively. The corresponding M_w were 39.0 kDa for PM6_C_1 and 42.2 kDa for PM6_C_2, with a precision range of ± 1.90 kDa. As shown in Supplementary Figs. 34 and Supplementary Table 15, all three batches of doped materials (including PM6_C, PM6_C_1 and PM6_C_2, respectively) enabled the PM6:Y6 binary system to achieve PCEs exceeding 15.5% at a preparation speed of 30 $m\ min^{-1}$ in the air. The comparable device efficiencies demonstrated the successful achievement of B2B synthesis of PM6 additives suitable for high-throughput manufacturing. In summary, the validation studies demonstrate the general applicability of the in-situ PL monitoring technique in D-A conjugated polymers with low to medium M_w s.”.

Figure 5 in the revised main text. The universality study results of the Y-series P_As and BDT-based polymers. (A) The molecular structures of the investigated P_A and P_D materials. The M_w s and corresponding PCEs of the PY-IT (B) and PYF-T-o (C) batches were investigated in the PM6-based all-polymer systems. (D) The M_w s of the PTIB batches. (E) The M_w s and corresponding PCEs of PM6 batches investigated in the high-speed printing systems.

Response to Referee #2:

Comments to the Author: *In this manuscript, Xu et al report the use of photoluminescence (PL) spectroscopy for in-situ monitoring of the degree of polymerization of polymer acceptors designed for organic photovoltaics (OPV) during its synthesis via Stille polymerization. For this polymer acceptor system (namely PYT), its performance in OPV devices has been previously found to depend strongly on its degree of polymerization/oligomerization, but currently there is a lack of method for real-time monitoring of the polymerization degree during its synthesis that leads to large batch-to-batch variations in device performance. The in-situ PL method reported in this manuscript tracks the change in (1) PL peak position, (2) FWHM of PL spectrum and (3) PL peak intensity as a function of time (minutes) during the polymerization process. Since these PL spectral parameters change with increasing polymerization degree, the authors were able to establish a quantitative model and achieve in-situ monitoring of polymerization over reaction time and controlled synthesis of PYT with varying molecular weights.*

Overall this manuscript reports a practical method for improving the synthesis procedure of polymers useful for all-polymer OPV devices, which is important for future research and development of this promising technology. However, I feel this manuscript may be more suitable for a more specialized journal as the results do not contain significant scientific novelty. Some technical comments for consideration are listed below.

Response: We highly appreciate the reviewer's insightful and helpful comments on our manuscript. We have provided the necessary revisions and descriptions in the revised manuscript. In the following, we give a point-by-point reply to your comments.

Additionally, we would like to further provide our viewpoints. The *in-situ* PL technology has not been studied for real-time monitoring of Stille cross-coupling polymerization, and real time, in-situ FTIR, Raman and Absorption Spectroscopy

cannot be used to monitor this type of polymerization reaction. We demonstrate the application and potential advantages of in-situ PL technology through our work, which is much important to improve the polymerization reactions of donor-acceptor fundamental polymer materials for all kinds of organic electronics. Additionally, in the revised manuscript, we showed this technique worked well for the other polymer donor and acceptor materials (including PY-OT, PYF-T-o, PM6 and PTIB) with suitable molecular weights, strongly demonstrating the approach's validity. Besides, the manuscript presents a new analytical technique with potential interdisciplinary applications, not just specialized results. Journals like Nat. Commun. aim to advance science across disciplines, which this work does by introducing in-situ PL monitoring of polymerization. In summary, the work's innovations and significance for cross-disciplinary science are obvious. We hope that introducing a new method for an important problem like polymerization reaction of conjugated polymers has wide-reaching implications worthy of a journal like Nat. Commun.

Comments:

Comment 1. *It may be helpful to discuss about the accuracy of this method in determining the material's molecular weight. How this accuracy may change vary between different types of polymers should also be discussed.*

Response: Thanks for the reviewer's suggestions. We add a discussion of the accuracy of the method in determining the molecular weight of PYT, as follows,

“We further analyzed the accuracy of the DP monitoring technology. As shown in Supplemental Fig. 16, the mean and standard deviation for the M_w of the ten PYT batches are 14.45 and 0.45, respectively, with an accuracy of 97%, which achieved DP precise customization of PYT.”

Additionally, as shown in Figure 5 in the revised main text (extracted from

Supplementary Figs. 19-34 and Supplementary Tables 9-15 in the revised SI file), the validation studies demonstrate the general applicability of the *in-situ* PL monitoring technique in D-A conjugated polymers with low to medium M_{w} s.

We added the corresponding descriptions in the section of “*Universalization of the DP monitoring technology*” in the revised main text, as follows:

“*Considering the effect of chemical structure modification on the spectra, we monitored the PL of regioregular (PY-IT⁴³ and PY-OT⁴³), fluorinated (PYF-T-o⁴⁴), BDT-based (PTIB⁴⁵) P_{AS} and BDT-based P_D (PM6⁴⁶) under polymerization process (the chemical structures are listed in **Fig. 5A**), to further explore the suitability and practicality of the DP monitoring technology. A new catalyst batch (Batch_C, **Supplemental Fig. 18**) was used for subsequent experiments. Similar to the process optimized for PYT, we controlled the reaction time to obtain polymers (batches A-E) for each system. The batch (X) with the optimal PCE then underwent interference experiments to synthesize batches X_1 and X_2, varying the catalyst amount, which are 1.33 and 0.67 times than that of X respectively.”;*

“*The 2D PL spectrum maps of PY-IT synthesized by polymerization of monomers Y5-C20-Br- γ and 2,5-bis(trimethylstannyl)thiophene are shown in **Supplemental Fig. 19**. Extracted PP, PI and PPC parameters versus time are presented in **Supplemental Fig. 20**, respectively. The three spectral parameters (PP, PI, and PPC) shifted from 796.6 to 786.3 nm for PP, 2.75×10^4 to 1.41×10^4 for PI, and 802.1 to 791.9 nm for PPC in the first 69 min (PY-IT_A), and then, the three spectral parameters rapidly increased to (825.383 nm, 4.99×10^4 and 831.234 nm for PY-IT_B and PY-IT_C), followed by a slow increase to (830.316 nm, 4.91×10^4 and 834.717 nm for PY-IT_D). Afterwards, these PL parameters reached stable as M_w increased from 5.7 to 19.0 kDa across PY-IT batches A-E (**Fig. 5B**, **Supplemental Fig. 21** and **Supplemental Table 9**). Device performance versus M_w established for PY-IT showed batch D with the highest PCE of 15.53%. Batches D_1 and D_2 synthesized while varying catalyst, which showed the reaction time of 113 and 267 mins, respectively, almost replicated batch PY-IT_D's M_w (12.4 and*

11.5 kDa) and PCE (15.44% and 15.42%) values (**Supplemental Table 9**, **Supplemental Table 10** and **Supplemental Fig. 22**). The results demonstrate the realization of the B2B synthesis of PY-IT using the developed DP monitoring technique.”;

“Additionally, we monitored the polymerization reaction of the PYT derivative PY-OT to better understand the effect of bromine isomerization at the terminal monomer groups. PY-OT was synthesized from monomers Y5-C20-Br- δ and 2,5-bis(trimethylstannyl)thiophene (**Supplemental Fig. 2**). As shown in **Supplemental Fig. 23**, the PL spectral parameters for PY-OT exhibited different trends compared to PY-IT. It can be found that the parameters began sharply decreasing after 100 minutes of reaction. Furthermore, some of the polymeric product precipitated out of solution when the reaction ceased, with a M_w of 30.0 kDa. This observation suggests the bromine atom located closer to the cyano group in the monomer displays enhanced reactivity, likely due to the stronger electron-withdrawing properties of the cyano group rendering the carbon-bromine bond more susceptible to nucleophilic palladium-catalyzed insertion. In short, monitoring the polymerization reactions of PY-IT and PY-OT provided insight into how bromine isomerization at the monomer terminus can influence the reaction kinetics and polymer properties.”;

“The effect of fluorine substitution on reactivity by monitoring the DP of a fluorinated PYT derivative, PYF-T-o, synthesized from monomers Y-OD-FBr-o (**Supplemental Fig. 2**) and 2,5-bis(trimethylstannyl)thiophene. The 2D PL spectrum maps of PYF-T-o polymerization process are listed in **Supplemental Fig. 24**, and corresponding spectral parameters are summarized in **Supplemental Fig. 25**. The M_w for batches PYF-T-o_A-E ranged from 7.3-20.7 kDa (**Fig. 5C**, **Supplemental Fig. 26** and **Supplemental Table II**). The PCEs of relevant devices for these batches blended with PM6 ranged from 7.91%-12.20% (**Fig. 5C**, **Supplemental Fig. 27** and **Supplemental Table 12**). Batch PYF-T-o_C was selected as a reference for verifying B2B synthesis. Note that batches PYF-T-o_C_1 and PYF-T-o_C_2 reached the set parameters ranges after 78 and 120

mins, respectively, with M_{ws} of 10.9 and 10.5 kDa, respectively. The PCEs of both devices based on PYF-T-o_C_1 and PYF-T-o_C_2 are approximately 12.0%. Moreover, fluorine substitution activated the neighboring bromine, allowing a M_n of 18.7 kDa after 170 minutes. However, it also enhanced aggregation-induced quenching compared to PY-IT, implying stronger F-S noncovalent interactions increase aggregation at high temperatures.”;

“We extended the in-situ PL monitoring approach to non-Y-series systems to further assess its generalizability, exploring B2B synthesis of the BDT-based P_A (PTIB⁴⁵, synthesized from BDT-TF-Sn and TIC-Br monomers (**Supplemental Fig. 2**)). The 2D PL spectral maps, extracted parameters (PP, PI, PPC), and GPC data are shown in **Supplementary Figs. 28-30** and **Supplementary Table 13**, respectively. Compared to Y-series polymers, PTIB exhibited weaker fluorescence at lower M_w (<3.6 kDa, batch PTIB_A) and significant increases in PL parameters until M_w reached ~9.3 kDa (batch PTIB_C). The parameters then slowly increased, stabilizing at (742.808 nm, 1.48×10^4 , 756.406 nm) for batch PTIB_D with a M_w of 13.2 kDa. In contrast, PP decreased for batch PTIB_E ($M_w > 13.2$ kDa), possibly due to increased backbone distortion for PTIB. Taking PTIB_C ($M_w \sim 9.3$ kDa, reaction time = 183 mins) as the reference, we further synthesized batches PTIB_C_1 and PTIB_C_2, which had M_{ws} of 9.4 and 8.8 kDa with the reaction time of 138 and 206 mins, respectively.”;

“Apart from the BDT-based P_A PTIB, we also selected commercially available D-A polymer donor, PM6, to illustrate the advantages of this in-situ PL technique. The detailed trends of relevant parameters as well as the corresponding M_{ws} of different batches are demonstrated in **Supplementary Figs. 31-33** and **Supplementary Table 14**, respectively. In accordance with our previous work,⁵¹ we conducted self-doping experiments using PM6 polymers of varying M_{ws} in the PM6:Y6 system. Our findings revealed that the medium M_w variant (PM6_C) significantly enhances the high-speed processing of the active layer, while having minimal impact on its device performance. Thus, further taking batch PM6_C as an example, we synthesized other two batches

PM6_C_1 and PM6_C_2, successfully attained the desired PL parameters range within 93 and 203 minutes, respectively. The corresponding M_w were 39.0 kDa for PM6_C_1 and 42.2 kDa for PM6_C_2, with a precision range of ± 1.90 kDa. As shown in Supplementary Figs. 34 and Supplementary Table 15, all three batches of doped materials (including PM6_C, PM6_C_1 and PM6_C_2, respectively) enabled the PM6:Y6 binary system to achieve PCEs exceeding 15.5% at a preparation speed of 30 m min^{-1} in the air. The comparable device efficiencies demonstrated the successful achievement of B2B synthesis of PM6 additives suitable for high-throughput manufacturing. In summary, the validation studies demonstrate the general applicability of the in-situ PL monitoring technique in D-A conjugated polymers with low to medium M_w s.”.

Figure 5 in the revised main text. The universality study results of the Y-series P_A s and BDT-based polymers. (A) The molecular structures of the investigated P_A and P_D materials. The M_w s and corresponding PCEs of the PY-IT (B) and PYF-T-o (C) batches were investigated in the PM6-based all-polymer systems. (D) The M_w s of the PTIB batches. (E) The M_w s and corresponding PCEs of PM6 batches investigated in the high-speed printing systems.

Comment_2. Fig 3G, H: units for time and weight are missing.

Response: Many thanks for pointing out this mistake. We corrected it as shown in Fig. 3G and 3H in the revised main text.

Response to Referee #3:

Comments to the Author: *The manuscript by Xu et al investigates the real-time monitoring of the oligomerization degree (OD) of polymer acceptors to acquire high-quality polymer acceptor (PYT) with no batch-to-batch (B2B) variations, which is benefit for fabricating high performance all-polymer solar cells. Herein, Xu et al develop an in-situ photoluminescence (PL) system to track and estimate the OD through the Stille polymerization. Moreover, they establish a mathematical model for the quantitative prediction of the experimental molecular weight and device efficiencies. As a result, the similar PCEs have been achieved through the monitor at different condition, which is promising and potential for the industrial scalability and desired cost reduction in the future. The publication of the manuscript in this prestigious journal seems to be appropriate if the questions and comments below are fully addressed.*

Response: Thank you for your positive comments. We have adopted all the suggestions, and trust that all of your comments have been addressed accordingly in the revised manuscript. In the following, we give a point-by-point reply to your excellent comments.

Comments:

Comment_1. *The M_{ws} of the three PYT batches are supposed to be provided in the article (Page 5, Line 135) and Figure 1c, which makes the comparison between different batches clearer.*

Response: Thanks a lot for the reviewer's suggestions. We added the M_{ws} of PYT_L, PYT_M and PYT_H in the revised main text (Figure 1C).

Comment_2. *The reaction temperature should be marked on the reaction arrow in Figure 1a. The name of Figure 1c seems to miss PM6 and PM7.*

Response: Many thanks for pointing out these mistakes. According to your suggestions, the reaction temperature has been labeled on the reaction arrow. In the name of Fig. 1C, we previously denoted the three polymer donors PBDB-T, PM6, and PM7 by PBDB-T derivatives, but this may be confusing, so we recorrected the corresponding description, as follows,

“Photovoltaic performance characteristics of all-PSCs fabricated from PYT with PBDB-T, PM6 and PM7 based on different M_{ws} .”

Comment_3. *Please explain why the peak intensity (PP) of PYT increases firstly and then weakens.*

Response: Thanks a lot for your thoughtful comment. The absorptivity and fluorescence intensity of P_{AS} are usually positively correlated with the degree of conjugation over a range of molecular weights. When the molecular weight reaches a certain level, skeleton rigidity and polymer chain entanglement inhibit the vibration and rotation of molecular skeleton and functional groups. Accordingly, the fluorescence intensity of the reaction system starts to decrease when the aggregation-induced fluorescence quenching effect strengthens over the conjugation-enhanced fluorescence effect, and this phenomenon is more significant in the PM6 polymerization process, which will be further discussed in Comment_7. Therefore, the peak intensity (PI) of PYT increases first and then weakens.

Comment_4. *Figure 3I seems to miss the arrows to verify the ownership of lines and vertical coordinate.*

Response: Many thanks for pointing out this mistake. We added the corresponding arrows in Figure 3I in the revised main text.

Comment_5. Please give detailed exposed time in the article because it appears in Figure 4c and Table 1.

Response: Thank you very much for your suggestion. The corresponding description is supplemented by the following: “Additionally, taking batch A provided by Supplier_1 as an example, we further investigated the correlations between the reactivity of the palladium(0) catalyst and the reaction time of PYT polymerization through oxidizing the catalyst in the air, *where catalysts Batch_A0, Batch_A3, Batch_A7 and Batch_A15 are batches of Supplier_1 exposed to air for 0, 3, 7 and 15 days, respectively* (Supplementary Fig. 16).”.

Comment_6. Please carefully check if the titles of the Figures and Tables are correct (including capitalization, punctuation, grammar, etc.)

Response: Thank you very much for your pertinent suggestions. The expressions have been modified and the language has been polished.

Comment_7. This work seems to lack universal validation. I wonder if this method is also suitable for the synthesis of other polymer acceptors (such as PAs with stereoregularity)?

Response: Thanks for the reviewer’s suggestion. In the revised manuscript, we further investigated the applicability of PL monitoring technology to Y-series P_A s with stereoregularity (PY-IT and PY-OT), Y-series P_A with fluoro-substitution (PYF-T-*o*), BDT-based P_A (PTIB) and BDT-based P_D (PM6). As shown in Figure 5 in the revised

main text (extracted from Supplementary Figs. 19-34 and Supplementary Tables 9-15 in the revised SI file), the validation studies demonstrate the general applicability of the *in-situ* PL monitoring technique in D-A conjugated polymers with low to medium $M_{w,s}$.

We added the corresponding descriptions in the section of “*Universalization of the DP monitoring technology*” in the revised main text, as follows:

“*Considering the effect of chemical structure modification on the spectra, we monitored the PL of regioregular (PY-IT⁴³ and PY-OT⁴³), fluorinated (PYF-T-o⁴⁴), BDT-based (PTIB⁴⁵) P_{AS} and BDT-based P_D (PM6⁴⁶) under polymerization process (the chemical structures are listed in **Fig. 5A**), to further explore the suitability and practicality of the DP monitoring technology. A new catalyst batch (Batch_C, **Supplemental Fig. 18**) was used for subsequent experiments. Similar to the process optimized for PYT, we controlled the reaction time to obtain polymers (batches A-E) for each system. The batch (X) with the optimal PCE then underwent interference experiments to synthesize batches X_1 and X_2, varying the catalyst amount, which are 1.33 and 0.67 times than that of X respectively.”;*

“*The 2D PL spectrum maps of PY-IT synthesized by polymerization of monomers Y5-C20-Br- γ and 2,5-bis(trimethylstannyl)thiophene are shown in **Supplemental Fig. 19**. Extracted PP, PI and PPC parameters versus time are presented in **Supplemental Fig. 20**, respectively. The three spectral parameters (PP, PI, and PPC) shifted from 796.6 to 786.3 nm for PP, 2.75×10^4 to 1.41×10^4 for PI, and 802.1 to 791.9 nm for PPC in the first 69 min (PY-IT_A), and then, the three spectral parameters rapidly increased to (825.383 nm, 4.99×10^4 and 831.234 nm for PY-IT_B and PY-IT_C), followed by a slow increase to (830.316 nm, 4.91×10^4 and 834.717 nm for PY-IT_D). Afterwards, these PL parameters reached stable as M_w increased from 5.7 to 19.0 kDa across PY-IT batches A-E (**Fig. 5B**, **Supplemental Fig. 21** and **Supplemental Table 9**). Device performance versus M_w established for PY-IT showed batch D with the highest PCE of 15.53%. Batches D_1 and D_2 synthesized while varying catalyst, which showed the reaction time of 113 and 267 mins, respectively, almost replicated batch PY-IT_D's M_w (12.4 and*

11.5 kDa) and PCE (15.44% and 15.42%) values (**Supplemental Table 9, Supplemental Table 10 and Supplemental Fig. 22**). The results demonstrate the realization of the B2B synthesis of PY-IT using the developed DP monitoring technique.”;

“Additionally, we monitored the polymerization reaction of the PYT derivative PY-OT to better understand the effect of bromine isomerization at the terminal monomer groups. PY-OT was synthesized from monomers Y5-C20-Br- δ and 2,5-bis(trimethylstannyl)thiophene (**Supplemental Fig. 2**). As shown in **Supplemental Fig. 23**, the PL spectral parameters for PY-OT exhibited different trends compared to PY-IT. It can be found that the parameters began sharply decreasing after 100 minutes of reaction. Furthermore, some of the polymeric product precipitated out of solution when the reaction ceased, with a M_w of 30.0 kDa. This observation suggests the bromine atom located closer to the cyano group in the monomer displays enhanced reactivity, likely due to the stronger electron-withdrawing properties of the cyano group rendering the carbon-bromine bond more susceptible to nucleophilic palladium-catalyzed insertion. In short, monitoring the polymerization reactions of PY-IT and PY-OT provided insight into how bromine isomerization at the monomer terminus can influence the reaction kinetics and polymer properties.”;

“The effect of fluorine substitution on reactivity by monitoring the DP of a fluorinated PYT derivative, PYF-T-o, synthesized from monomers Y-OD-FBr-o (**Supplemental Fig. 2**) and 2,5-bis(trimethylstannyl)thiophene. The 2D PL spectrum maps of PYF-T-o polymerization process are listed in **Supplemental Fig. 24**, and corresponding spectral parameters are summarized in **Supplemental Fig. 25**. The M_w for batches PYF-T-o_A-E ranged from 7.3-20.7 kDa (**Fig. 5C, Supplemental Fig. 26 and Supplemental Table II**). The PCEs of relevant devices for these batches blended with PM6 ranged from 7.91%-12.20% (**Fig. 5C, Supplemental Fig. 27 and Supplemental Table 12**). Batch PYF-T-o_C was selected as a reference for verifying B2B synthesis. Note that batches PYF-T-o_C_1 and PYF-T-o_C_2 reached the set parameters ranges after 78 and 120

mins, respectively, with M_{ws} of 10.9 and 10.5 kDa, respectively. The PCEs of both devices based on PYF-T-o_C_1 and PYF-T-o_C_2 are approximately 12.0%. Moreover, fluorine substitution activated the neighboring bromine, allowing a M_n of 18.7 kDa after 170 minutes. However, it also enhanced aggregation-induced quenching compared to PY-IT, implying stronger F-S noncovalent interactions increase aggregation at high temperatures.”;

“We extended the in-situ PL monitoring approach to non-Y-series systems to further assess its generalizability, exploring B2B synthesis of the BDT-based P_A (PTIB⁴⁵, synthesized from BDT-TF-Sn and TIC-Br monomers (**Supplemental Fig. 2**)). The 2D PL spectral maps, extracted parameters (PP, PI, PPC), and GPC data are shown in **Supplementary Figs. 28-30** and **Supplementary Table 13**, respectively. Compared to Y-series polymers, PTIB exhibited weaker fluorescence at lower M_w (<3.6 kDa, batch PTIB_A) and significant increases in PL parameters until M_w reached ~9.3 kDa (batch PTIB_C). The parameters then slowly increased, stabilizing at (742.808 nm, 1.48×10^4 , 756.406 nm) for batch PTIB_D with a M_w of 13.2 kDa. In contrast, PP decreased for batch PTIB_E ($M_w > 13.2$ kDa), possibly due to increased backbone distortion for PTIB. Taking PTIB_C ($M_w \sim 9.3$ kDa, reaction time = 183 mins) as the reference, we further synthesized batches PTIB_C_1 and PTIB_C_2, which had M_{ws} of 9.4 and 8.8 kDa with the reaction time of 138 and 206 mins, respectively.”;

“Apart from the BDT-based P_A PTIB, we also selected commercially available D-A polymer donor, PM6, to illustrate the advantages of this in-situ PL technique. The detailed trends of relevant parameters as well as the corresponding M_{ws} of different batches are demonstrated in **Supplementary Figs. 31-33** and **Supplementary Table 14**, respectively. In accordance with our previous work,⁵¹ we conducted self-doping experiments using PM6 polymers of varying M_{ws} in the PM6:Y6 system. Our findings revealed that the medium M_w variant (PM6_C) significantly enhances the high-speed processing of the active layer, while having minimal impact on its device performance. Thus, further taking batch PM6_C as an example, we synthesized other two batches

PM6_C_1 and PM6_C_2, successfully attained the desired PL parameters range within 93 and 203 minutes, respectively. The corresponding M_w were 39.0 kDa for PM6_C_1 and 42.2 kDa for PM6_C_2, with a precision range of ± 1.90 kDa. As shown in Supplementary Figs. 34 and Supplementary Table 15, all three batches of doped materials (including PM6_C, PM6_C_1 and PM6_C_2, respectively) enabled the PM6:Y6 binary system to achieve PCEs exceeding 15.5% at a preparation speed of 30 m min^{-1} in the air. The comparable device efficiencies demonstrated the successful achievement of B2B synthesis of PM6 additives suitable for high-throughput manufacturing. In summary, the validation studies demonstrate the general applicability of the in-situ PL monitoring technique in D-A conjugated polymers with low to medium M_w 's.

Figure 5 in the revised main text. The universality study results of the Y-series P_A s and BDT-based polymers. (A) The molecular structures of the investigated P_A and P_D materials. The M_w s and corresponding PCEs of the PY-IT (B) and PYF-T-o (C) batches were investigated in the PM6-based all-polymer systems. (D) The M_w s of the PTIB batches. (E) The M_w s and corresponding PCEs of PM6 batches investigated in the high-speed printing systems.

REVIEWER COMMENTS

Reviewer #1 (Remarks to the Author):

The authors have thoroughly explained the differences between their work and other monitoring techniques of polymerization. In addition, they also carried out additional experiments to validate the universal applicability of their polymerization monitoring technique. Therefore, I think the manuscript can be accepted for publication in Nature Communications.

Reviewer #2 (Remarks to the Author):

In the revised manuscript, the authors have fixed technical issues raised by the reviewers and included additional data to show that in-situ PL can also be used to monitor the degree of polymerization in a number of OPV donor and acceptor polymers during synthesis. By calibrating the in-situ PL results with standard GPC parameters (Mw), this method can help to speed up the search for the optimal polymer synthesis conditions and help maintain batch quality. However, I stand by my view that this work does not provide significant scientific insights and should be published in a more specialized journal instead of Nature Communications.

Reviewer #3 (Remarks to the Author):

We acknowledge that the corrections made have led to significant improvements in addressing the raised questions. However, we recognize that there are still important issues that need to be resolved before the article can be considered for further publication.

1. It is important to include the synthesis introduction of high-efficiency donor polymers such as PM6, D18, PBQx-T-F, etc., in the second paragraph. This addition will provide comprehensive background information and establish connections between the article and relevant studies in the field.
2. Regarding the authors' proposition on the importance of in-situ monitoring and the lack of scientific investigation, it would be beneficial to provide further scientific explanation at appropriate points.
3. Considering the widespread utilization of in-situ UV-vis technology in the field of organic photovoltaics, it is advisable to suggest including in-situ UV information as supporting material. Additionally, the authors should explain why they chose PL measurements over UV-vis as the monitoring method. This will help readers better understand the authors' choices and research approach.

4. Given the significance of PY-IT in high-efficiency all-polymer solar cells, it is recommended to emphasize the investigation of PY-IT in the article. This includes discussing its synthesis, performance, and applications. Such focus will align the article with current research trends and applications.
5. With regard to the issue of the molecular weight of PYTM (10.6 kDa) being very close to PYTL (9.7 kDa), it would be beneficial to suggest that the authors provide a higher molecular weight batch of PYTM (around 1.3 kDa). This will enable a more comprehensive study of the effect of molecular weight on performance and enhance the understanding of the structure-property relationship of the polymers.
6. Regarding the lower efficiency achieved in PY-IT-based all-polymer solar cells (only reaching 15.4%), it would be helpful to propose a comparative analysis to explain why the efficiency in this study is lower than recent works that achieved over 17% efficiency (Adv. Mater. 2023, 2306990; Adv. Mater. 2023, 2308061, etc.). This analysis may involve material optimization, device structure improvements, and other relevant factors.
7. Regarding the synthesis of PTIB mentioned in the article without discussing its photovoltaic performance, it would be appropriate to inquire about the reasons behind synthesizing PTIB and excluding its photovoltaic performance study. This can prompt the authors to provide additional information on the synthesis and potential applications of PTIB.

Response to Referee #1:

Comments to the Author: *The authors have thoroughly explained the differences between their work and other monitoring techniques of polymerization. In addition, they also carried out additional experiments to validate the universal applicability of their polymerization monitoring technique. Therefore, I think the manuscript can be accepted for publication in Nature Communications.*

Response: We appreciate the reviewer for supporting this research to be published in Nature Communications again. Many thanks.

Response to Referee #2:

Comments to the Author: *In the revised manuscript, the authors have fixed technical issues raised by the reviewers and included additional data to show that in-situ PL can also be used to monitor the degree of polymerization in a number of OPV donor and acceptor polymers during synthesis. By calibrating the in-situ PL results with standard GPC parameters (M_w), this method can help to speed up the search for the optimal polymer synthesis conditions and help maintain batch quality. However, I stand by my view that this work does not provide significant scientific insights and should be published in a more specialized journal instead of Nature Communications.*

Response: We are grateful to the reviewer for acknowledging our additional revisions. We agree with the reviewer's opinion that we have fixed some technical issues, which can help to speed up the search for the optimal polymer synthesis conditions and help maintain batch quality. Moreover, we also believe that our work provides significant scientific insights. In our opinion, there are topics or works dedicated to the discovery and invention of new detector technologies and new characterization tools, and the new technologies and new tools to be invented by such topics are the scientific problems of the project, and how to invent this new technology and new tools is the technical problems of the subject. In this work, both scientific and technical problems have been conducted to solve the key issue of the batch-to-batch variations in device performance.

In the strict sense, the invention and discovery of new technologies and tools are not fundamental science problems, but applied science problems. But from the perspective of solving consistency issues of device performance in any organic electronics, the invention and discovery of new technologies and new tools should also be supported. This innovation, as detailed in our work, not only enhances the overall quality of the resulting polymers but also paves the way for their commercial application. We firmly believe that if our innovative research could be published in a high-impact journal like *Nat Commun*, it can further promote the rapid development of organic electronics.

Response to Referee #3:

Comments to the Author: *We acknowledge that the corrections made have led to significant improvements in addressing the raised questions. However, we recognize that there are still important issues that need to be resolved before the article can be considered for further publication.*

Response: We express our gratitude to the reviewer for acknowledging our revised manuscript and providing valuable suggestions for improvement. Here we give a point-by-point reply to your comments.

Comment 1. *It is important to include the synthesis introduction of high-efficiency donor polymers such as PM6, D18, PBQx-T-F, etc., in the second paragraph. This addition will provide comprehensive background information and establish connections between the article and relevant studies in the field.*

Response: Thanks for the reviewer's suggestion. We added the synthesis introduction of high-efficiency polymer donors in the second paragraph, as follows:

“Moreover, Hwang and his coworkers investigated and synthesized PTB7 in a short reaction time with a specific molecular mass through a rapid-flow synthesis system.³⁶ Smeets et al. adopted a combination of Buchwald catalyst and droplet-flow chemistry to minimize B2B variations and structural defects of high-efficiency polymer donors PM6 and D18.³⁷ Despite this, recent synthesis experience tells us that even under the same reaction conditions and process protocols, there is usually no guarantee that the M_w and \mathcal{D} values of the conjugated polymers obtained each time are comparable.”

Comment 2. *Regarding the authors' proposition on the importance of in-situ monitoring and the lack of scientific investigation, it would be beneficial to provide*

further scientific explanation at appropriate points.

Response: Thanks for the reviewer's suggestion. We provided the relevant scientific explanations at appropriate points in the main text, as follows:

“However, the aromaticity and reactivity of conjugated polymers undergo variations with the chain length and pre-aggregation among the living polymer chains during chain-growth polymerization. Consequently, solving the practical dilemma of polymer batch-to-batch (B2B) variations through molecular structure regulation is often insurmountable.^{14, 24, 25, 27, 28, 29} To this end, achieving high precision tailored DPs remains an important topic of high interest for investigating material properties and improving device performances.”;

“This illustrates that both in-situ absorption and PL spectroscopy can provide possible pathways to obtain specific DP of a conjugated polymer in an offline manner during the polymerization reaction. However, in-situ PL technology possesses the irreplaceable advantages of simplicity, reliability, and sensitivity over absorption, FTIR, and Raman spectroscopy for organic semiconductor materials. This is because the absorptivity and fluorescence intensity of P_{AS} are usually positively correlated with the degree of conjugation over a range of molecular weights. When the molecular weight reaches a certain level, skeleton rigidity and polymer chain entanglement inhibit the vibration and rotation of molecular skeleton and functional groups. In this case, the emission spectra monitored by PL technology can dynamically respond to changes in the degree of radiative and non-radiative quenching of exciton online, providing a sensitive reflection of the physical and chemical state of the polymer chain compared to the other spectral techniques such as absorption, FTIR and Raman spectroscopy.”.

Comment_3. *Considering the widespread utilization of in-situ UV-vis technology in the field of organic photovoltaics, it is advisable to suggest including in-situ UV information as supporting material. Additionally, the authors should explain why they*

chose PL measurements over UV-vis as the monitoring method. This will help readers better understand the authors' choices and research approach.

Response: Thank you for the reviewer's comments. We supplement the *in-situ* UV-vis spectra of the PYT polymerization system (**Figure R1**, Supplementary Fig. 6 in the revised supplementary information). Owing to the high solution concentration of oligomers during PYT polymerization that results in absorbance beyond the range of instrumental measurements, it is difficult to evaluate the oligomerization degree by real-time monitoring of absorption spectra. In addition, we have further elaborated on the advantages of utilizing *in-situ* PL technology as the monitoring method over *in-situ* UV-vis spectroscopy and other spectral technologies in the main text based on your suggestion, as follows:

“This illustrates that both in-situ absorption and PL spectroscopy can provide possible pathways to obtain specific DP of a conjugated polymer in an offline manner during the polymerization reaction. However, in-situ PL technology possesses the irreplaceable advantages of simplicity, reliability, and sensitivity over absorption, FTIR, and Raman spectroscopy for organic semiconductor materials. This is because the absorptivity and fluorescence intensity of P_{AS} are usually positively correlated with the degree of conjugation over a range of molecular weights. When the molecular weight reaches a certain level, skeleton rigidity and polymer chain entanglement inhibit the vibration and rotation of molecular skeleton and functional groups. In this case, the emission spectra monitored by PL technology can dynamically respond to changes in the degree of radiative and non-radiative quenching of exciton online, providing a sensitive reflection of the physical and chemical state of the polymer chain compared to the other spectral techniques such as absorption, FTIR and Raman spectroscopy.”

“As an example, Supplementary Fig. 5 shows the measured absorption coefficients of the corresponding PYT batches in chloroform-diluted solutions with the same concentration. As compared to PYT_L and PYT_M, PYT_H exhibited a higher absorption

coefficient, originating from the low π - π^* excitation energy and high oscillator strength.¹⁹ Owing to the high solution concentration of oligomers as well as the magnetic stirring process during PYT polymerization, it is not feasible to evaluate the oligomerization degree by real-time monitoring of absorption spectra (Supplementary Fig. 6).”.

Figure R1 (Supplementary Fig. 6 in the revised supplementary information). UV-vis absorption spectra of PYT polymerization system tested by an *in-situ* UV-vis setup. Note that absorbance beyond the range of instrumental measurements.

Comment 4. Given the significance of PY-IT in high-efficiency all-polymer solar cells, it is recommended to emphasize the investigation of PY-IT in the article. This includes discussing its synthesis, performance, and applications. Such focus will align the article with current research trends and applications.

Response: Thank you for the reviewer’s suggestion. We fully recognize the paramount importance of PY-IT in the current research landscape of all-polymer solar cells. With this understanding, we have delved into the *in-situ* PL technique, which has demonstrated promising results for PY-IT in our previous revised manuscript. Furthermore, in our previous work (*Joule* 4, 1070–1086, ref. 14; *J. Mater. Chem. C* **10**, 1850 ref. 27), we conducted a comprehensive exploration of the relationship between the molecular weight of the Y-series polymer acceptor and its resulting performance properties. These in-depth analyses allowed us to gain a deeper understanding of how

the structural characteristics of the polymer acceptor impact the overall performance.

Additionally, including an extensive discussion on the synthesis, performance, and applications of PY-IT might render the article more verbose. The current focus of this paper is to present a practical tool for the customization of suitable low-DP conjugated polymers in PSCs, thus warrants a concise and direct presentation. Our primary objective is to maintain the clarity and brevity of the article while effectively communicating our research findings and methodology, so we wish to maintain the current results. We acknowledge the importance of discussing the synthesis, performance, and applications of PY-IT in the broader context of multi-scale studies. However, this motivation is beyond the scope of this work. Recently, we start to utilize PY-IT polymer materials and will combine relevant developed and/or introduced machine learning algorithms and previously collected measurements to perform statistical correlation analysis of molecular weight and polydispersity index parameters. Based on the above PY-IT test results and the introduction of more reaction conditions and external factors, we will combine the online aggregation degree monitoring device and machine learning tools to evaluate and screen out the algorithm model with higher accuracy, especially in terms of polydispersity index parameter. Additionally, we will use the molecular weight evaluation platform to achieve macro preparation of high-performance PY-IT materials and explore their practical application potential. Thus, we believe that this new work, whose motivation is obviously different with the recent investigations, can further overcome the batch-to-batch variations in device performance and facilitate the development of relevant organic electronics.

***Comment_5.** With regard to the issue of the molecular weight of PYTM (10.6 kDa) being very close to PYTL (9.7 kDa), it would be beneficial to suggest that the authors provide a higher molecular weight batch of PYTM (around 1.3 kDa). This will enable a more comprehensive study of the effect of molecular weight on performance and enhance the understanding of the structure-property relationship of the polymers.*

Response: Thank you for raising the concern regarding the molecular weights of PYT_L and PYT_M. The polymer acceptor materials, PYT_L, PYT_M and PYT_H, originate from a portion of our prior research published in *Joule* 4, 1070–1086 (ref. 14). That study delved into the structure-property relationship between molecular weight and performance of polymer acceptors in all-polymer systems. It is worth noting that although the molecular weights of PYT_L and PYT_M are closely related, their molecular properties (PL peak positions of 802 nm and 814 nm, respectively) and photovoltaic performances (PCE of 13.64% and 12.92%, respectively, for the PM6-based devices) show considerable differences. These observed differences further emphasize the importance of developing DP monitoring technology. Furthermore, in the subsequent sections of this work (Data analysis for model building), we have fully demonstrated the effect of various PYT molecular weights on performance, which will deepen the readers' understanding of the structure-property relationship of the polymers. Therefore, we would like to retain the molecular weight of PYT_M as it appears to be more rational choice given the context of this study. Many thanks again for your suggestions and understanding.

Comment_6. Regarding the lower efficiency achieved in PY-IT-based all-polymer solar cells (only reaching 15.4%), it would be helpful to propose a comparative analysis to explain why the efficiency in this study is lower than recent works that achieved over 17% efficiency (Adv. Mater. 2023, 2306990; Adv. Mater. 2023, 2308061, etc.). This analysis may involve material optimization, device structure improvements, and other relevant factors.

Response: Thanks very much for the reviewer's comment. The relative lower efficiency of the PY-IT-based device in this study compared to recent work may be attributed to a pivotal factor: we only took methanol, acetone and *n*-hexane as eluents to purify the crude product, in order to reduce effect on the M_w results. The absence of dichloromethane as an eluent might have resulted in a larger \bar{D} of the polymer. Apart from the purification conditions, other processing conditions (like glovebox atmosphere

and temperature) and original materials (e.g., buffer layers and PM6 batches) may also significantly influence the device performance achieved by ourselves and other labs. We have included the relevant explanation in the main text as follows:

“The whole workflow for the automatic DP-monitoring technology is shown in Fig. 2d. In this workflow, to establish an effective model for a specific polymer, the products of multiple polymerizations were separately collected in the collecting chamber filled with methanol and further purified by methanol, acetone, and n-hexane. Subsequently, the trichloromethane fraction was concentrated to yield the resulting polymer. To minimize the interference of molecular weight by the purification operation, dichloromethane was not added as an eluent to further narrow the \bar{D} of the polymers, which may result in lower performance than reported in the literature.”.

Comment_7. *Regarding the synthesis of PTIB mentioned in the article without discussing its photovoltaic performance, it would be appropriate to inquire about the reasons behind synthesizing PTIB and excluding its photovoltaic performance study. This can prompt the authors to provide additional information on the synthesis and potential applications of PTIB.*

Response: Thank you to the reviewer for the comment. Following your suggestion, we added the relevant description in the main text, as follows,

“We extended the in-situ PL monitoring approach to non-Y-series systems to further assess its generalizability, exploring B2B synthesis of the BDT-based P_A PTIB⁴⁵ (synthesized from BDT-TF-Sn and TIC-Br monomers (Supplementary Fig. 2)). The polymer possesses lower synthetic complexity and molecular weight than the reported high-performance D-A type P_{AS} , while demonstrating significant potential for scale-up fabrication.”.